

# An energy budget approach to understand the Arctic warming during the Last Interglacial

Marie Sicard[1], Masa Kageyama[1], Sylvie Charbit[1], Pascale Braconnot[1], and Jean-Baptiste Madeleine[2]

[1]Laboratoire des Sciences du Climat et de l'Environnement, Institut Pierre Simon Laplace, Université Paris-Saclay, 91191 Gif-sur-Yvette Cedex, France
[2]Laboratoire de Météorologie Dynamique, Institut Pierre Simon Laplace, Sorbonne Université, 75252 Paris Cedex 05, France

**Correspondence:** Marie Sicard (marie.sicard@lsce.ipsl.fr)

**Abstract.** The Last Interglacial period (129-116 ka BP) is characterized by a strong orbital forcing which leads to a different seasonal and latitudinal distribution of insolation compared to the pre-industrial period. In particular, these changes amplify the seasonality of the insolation in the high latitudes of the northern hemisphere. Here, we investigate the Arctic climate response to this forcing by comparing the CMIP6 *lig127k* and *pi-Control* simulations performed with the IPSL-CM6A-LR model. Using

an energy budget framework, we analyse the interactions between the atmosphere, ocean, sea ice and continents.

In summer, the insolation anomaly reaches its maximum and causes a near-surface air temperature rise of 3.2°C over the Arctic region. This warming is primarily due to a strong positive surface downwelling shortwave radiation anomaly over continental surfaces, followed by large heat transfers from the continents back to the atmosphere. The surface layers of the Arctic Ocean also receives more energy, but in smaller quantity than the continents due to a cloud negative feedback. Furthermore, while

heat exchanges from the continental surfaces towards the atmosphere are strengthened, the ocean absorbs and stores the heat excess due to a decline in sea ice cover.

However, the maximum near-surface air temperature anomaly does not peak in summer like insolation, but occurs in autumn with a temperature increase of 4.0°C relative to the pre-industrial period. This strong warming is driven by a positive anomaly of longwave radiations over the Arctic ocean enhanced by a positive cloud feedback. It is also favoured by the summer and

autumn Arctic sea ice retreat ($-1.9 \times 10^6$ and $-3.4 \times 10^6$ km$^2$ respectively), which exposes the warm oceanic surface and allows heat stored by the ocean in summer and water vapour to be released. This study highlights the crucial role of the sea ice cover variations, the Arctic ocean, as well as changes in polar clouds optical properties on the Last Interglacial Arctic warming.

## 1   Introduction

In recent years, the Arctic climate system has been undergoing profound changes. Surface air temperature has increased by more than twice the global average over the last two decades (Meredith et al., 2019). This phenomenon, also known as the Arctic amplification, results from complex and numerous interactions involving the atmosphere, land surfaces, ocean and



cryosphere (Goosse et al., 2018). Sea ice loss is often cited as one of the main drivers of the Arctic amplification (Serreze and Barry, 2011). Over the past few decades, sea ice cover has responded very quickly to temperature fluctuations. Recent
satellite observations reveal large sea ice retreat in September peaking at –12.8 ± 2.3% per decade (relative to 1981–2010 mean; (Meredith et al., 2019)). A striking example is the minimum sea ice extent of 3.74 million km$^2$ reported in September 2020 by the NASA Earth Observatory which is the second lowest minimum since the beginning of satellite observations in 1979. During winter months, sea ice changes are smaller (about –2.7% ± 0.5% per decade in March; (Meredith et al., 2019)) but attest to a delayed start of the freezing season. Sea ice cover variations modify the albedo and affect the vertical exchanges
of heat and water vapor at the atmosphere-ocean interface. These albedo and water vapor effects has been previously analysed in a context of insolation variations, during the last interglacial–glacial transition (Khodri et al., 2005) and the mid-Holocene (Yoshimori and Suzuki, 2019). They also alter the density of oceanic water masses through salt rejection during the ice growing phase or through freshwater release during the melting period, and thereby, have the potential to modify the Atlantic Meridional Ocean Circulation (AMOC).

Other studies also highlight the important role of clouds and total water vapour content in amplifying or dampening the Arctic warming (Vavrus, 2004; Shupe and Intrieri, 2004; Vavrus et al., 2009; Graversen and Wang, 2009; Kay et al., 2016) . In particular, changes in the amount and characteristics of low-level clouds, ie. clouds occurring below 2000 m, strongly modulate the shortwave and longwave radiative budgets (Matus and L'Ecuyer, 2017). Remote sensing observations of clouds have shown the importance of cloud partitioning between liquid and ice phases on shortwave radiations received at the Earth's
surface (Cesana et al., 2012; Morrison et al., 2011). For a given water content, liquid water droplets are smaller and more abundant than their frozen counterparts. Their structural properties make them more efficient in reflecting incoming solar radiations back to space than ice crystals, which results in a cooling of the surface and the lowest layers of the atmosphere. Moreover, increased low-level cloud cover blocks longwave radiations and, thus, warm the atmosphere. This effect is more pronounced over newly open waters, where the enhanced moisture flux to the atmosphere contributes to the formation of low-
level clouds and leads to enhanced sea ice melt (Palm et al., 2010). These processes are now better captured by climate models, but cloud feedbacks remain a large source of uncertainty in climate projections (Flato et al., 2013; Ceppi et al., 2017). An other process also contributing to changes in temperature and humidity is the transport of heat and water vapor from low latitudes to the Arctic region (Khodri et al., 2003; Hwang et al., 2011; van der Linden et al., 2019). All these factors interact with each other and make it difficult to understand polar amplification (Serreze and Barry, 2011; Goosse et al., 2018).

The Arctic region experienced climatic variations in the past. Investigating past Arctic climate changes could therefore help to better understand processes involved in the Arctic amplification. Past interglacial periods are relevant examples for testing key dynamical processes and feedbacks under temperatures comparable to or warmer than present-day period. Because of the availability of numerous climate reconstructions, the Last Interglacial period, spanning from 129 to 116 years before present (BP), has been frequently used. It provides a good testing ground to clarify the relative importance and the cumulative effect of
the above mentioned processes. This period is characterized by a strong orbital forcing resulting in a global warming about 2°C at the peak warmth compared to the pre-industrial period (Turney and Jones, 2010; McKay et al., 2011; Capron et al., 2014). This warming is more pronounced in the high latitudes of the northern hemisphere. For the 127 ka period, paleodata suggest a





summer sea surface temperature increase of 1.1 ± 0.7 °C in the North Atlantic compared to the pre-industrial period (Capron et al., 2014, 2017) associated with huge variations of the cryosphere (here, sea ice and ice sheets). In their chronological framework, Thomas et al. (2020) demonstrate that the cryosphere responded early in the Last Interglacial period to the orbital forcing. The sea ice decline started as early as 130 ka and was followed by a retreat of the Greenland ice sheet starting around 128 ka. Although marine records agree on a significant decline in Arctic sea ice cover during the Last Interglacial period (Brigham-Grette and Hopkins, 1995; Nørgaard-Pedersen et al., 2007; Adler et al., 2009; Stein et al., 2017; Malmierca-Vallet et al., 2018; Kageyama et al., 2021), the presence of perennial or seasonal sea ice cover over the central Arctic Basin is still debated. Among CMIP6 climate models that have run Last Interglacial simulation, only two of them attest to summer ice-free conditions (Kageyama et al., 2021; Guarino et al., 2020).

In addition to Arctic sea ice loss, the Last Interglacial period is characterized by a retreat of both Greenland and the Antarctic ice sheets which have contributed to a sea level rise from 6 to 9 m (Kopp et al., 2009; Dutton et al., 2015).

Paleorecords indicate that greenhouse gas concentrations at 127 ka were similar to the pre-industrial ones. The main difference between the Last Interglacial and the pre-industrial forcings result from the astronomical forcing. Changes in obliquity and climatic precession affect both seasonal and annual solar radiation received at the top of the atmosphere. In the high latitudes of the northern hemisphere, the strong astronomical forcing leads to increased summer insolation at the top of the atmosphere peaking in June at more than 60 W m$^{-2}$ (fig. 1).

The aim of the study is to better understand how this insolation anomaly alters surface conditions and modifies the Arctic energy budget. To address this issue, we use the outputs of the IPSL-CM6A-LR global climate model to compare energy budget in the Arctic during the Last Interglacial and pre-industrial periods in the coupled atmosphere-ocean-land-sea ice system. The paper is organised as follows. Section 2 describes the IPSL-CM6A-LR model and the experimental design of the pre-industrial and Last Interglacial simulations. In section 3, we analyse the processes involved in the Arctic energy budget in order to determine their relative importance in the Arctic warming. We essentially focus on the summer and the autumn seasons during which the temperature rise is the highest. Section 4 discusses how model biases could influence our results.

## 2 Model and methods

### 2.1 The IPSL-CM6A-LR global model

The simulations analysed in this study were carried out using the IPSL-CM6A-LR model (Boucher et al., 2020). IPSL-CM6A-LR is a Global Climate Model (GCM) developed at Institut Pierre-Simon Laplace (IPSL). It includes three main components : the atmosphere (LMDZ, Hourdin et al. 2020), the ocean and sea ice (NEMO, Madec et al. 2019) and the land surface (ORCHIDEE, Krinner et al. 2005; Cheruy et al. 2020).

The horizontal resolution of the atmospheric model is 144x143 points in longitude and latitude corresponding to 157-km grid resolution at the equator. There are 79 vertical levels reaching 1.5 Pa at the top of the atmosphere. However, we use global fields interpolated on the 19 standard pressure levels defined for CMIP6 (Juckes et al., 2020). The ocean model has a resolution of 1° × 1°and 75 vertical layers. It includes a representation of sea ice (LIM, Vancoppenolle et al. 2008) and geochemistry (PISCES,





Aumont et al. 2015). The vegetation model ORCHIDEE uses fractions of 15 different plant functional types. Over ice-free areas, a 3-layer explicit snow model is also implemented, whereas, over ice sheets and glaciers, the snowpack is represented as a 1-layer snow scheme. Finally, the ORCHIDEE model also includes a carbon cycle representation, which implies that event though vegetation type are prescribed in each grid box the seasonal evolution of the leaf area index is computed. The horizontal
resolution of ORCHIDEE is the same as for the atmospheric component.

In this study, we use two simulations run as a part of the contribution of the fourth phase of Paleoclimate Modelling Intercomparison Project (PMIP4, Kageyama et al. 2018) of the Coupled Model Intercomparison Project CMIP6 (Eyring et al., 2016). The *piControl* experiment for 1850 CE, described in (Boucher et al., 2020) (following the CMIP6 protocol in Eyring et al. (2016)), is considered as our reference simulation and is cited as PI hereafter. The *lig127k* experiment, herafter the LIG
experiment, is a time-slice experiment corresponding to the 127 ka conditions following the PMIP4 protocol (Otto-Bliesner et al., 2017). As mentionned above, atmospheric $CO_2$ and other greenhouse gas (GHG) concentrations are close to their PI values and do not represent the main driver of the LIG climate. The LIG GHG concentrations are provided by Antarctic ice cores (Bereiter et al. 2015, Schneider et al. 2013 for $CO_2$; Loulergue et al. 2008 and Schilt et al. 2010a for $CH_4$ and Schilt et al. 2010a, b for $NO_2$) aligned on the AICC2012 chronology (Bazin et al., 2013). The Earth's astronomical parameters are
prescribed following Berger and Loutre (1991). In all simulations, the vernal equinox is fixed to March 21 at noon. Other boundary conditions such as paleogeography, ice sheet geometry or aerosols are the same as in PI simulation (for more details see Otto-Bliesner et al. 2017). Forcings and boundary conditions of both simulations are summarized in table 1.

The Last Interglacials simulation was initialised as the mid-Holocene one (Braconnot et al., 2021). The initial state is the year 1850 (1$^{st}$ January) of the CMIP6 reference pre-industrial simulation with the same model version (Boucher et al., 2020). The
model was then run for 350 years, which is longenough to bring the climate to equilibrium for Last Interglacial conditions. The last state of this simulation is used to initialise the reference PMIP4-CMIP6 *lig127k* simulation. The *lig127k* experiment was then run for 550 years, the last 50 years having high-frequency outputs for the analyses of extremes or to provide the boundary conditions for future regional simulations. This reference simulation is called CMIP6.PMIP.IPSL.IPSL-CM6A-LR.lig127k.r1i1p1f1 in the ESGF database.

Because of the combined effect of eccentricity and precession changes, the length of seasons relative to the insolation forcing is different between the LIG and the PI periods. However, both simulations use a fixed present-day calendar to compute online monthly averages, which is aligned with our current definition of season in terms of number of days for each months. As a consequence, this adds artificial biases in the analysis due to phase lag in the seasonal cycle, especially in boreal autumn (Joussaume and Braconnot, 1997; Timm et al., 2008; Bartlein and Shafer, 2019). To prevent from this "paleo-calendar effects",
we have adjusted the LIG monthly outputs with the PaleoCalAdjust algorithm (Bartlein and Shafer, 2019).

## 2.2  Climatological evaluation of IPSL-CM6A-LR for the Arctic region

The present-day climate simulated by the IPSL-CM6A-LR model has been evaluated in Boucher et al. (2020). Since the IPSL-CM5 model versions, significant improvements have been made for the turbulence, convection and clouds parameterizations (Hourdin et al., 2020; Madeleine et al., 2020). The adjustment of the subgrid-scale orography parameters has helped to cor-




rect a systematic bias in the representation of the Arctic sea ice (Gastineau et al., 2020). On an annual basis, this results in a general reduction of temperature biases from IPSL-CM5A-LR to IPSL-CM6A-LR versions (Boucher et al., 2020). In the high latitudes of the northern hemisphere, the cold bias in surface air temperature has been considerably reduced over the North Atlantic Ocean, as well as the warm bias over northern Canada. However, surface air temperatures are still too low over the Greenland ice sheet and a warm bias is also simulated in winter over the Arctic. The latter is associated with an underestimation

of the sea ice extent also found in summer. Despite these biases, the sea ice cover simulated with IPSL-CM6A-LR is in a better agreement with satellite data compared to previous model versions.

  The coupled model also tends to underestimate deep water formation in the North Atlantic and associated overturning circulation. In the northern hemisphere, the northward heat transport is more intense compared to previous versions, but remains weaker than that deduced from the few available direct observations. As the warm Atlantic and Pacific waters entering in the

Arctic basin affect the position of the sea ice front, they may be partly responsible for temperature and sea ice biases mentioned above.

    To evaluate the ability of IPSL-CM6A-LR to simulate the Last Interglacial climate, we compare the simulated surface temperature changes with the new data synthesis provided by Otto-Bliesner et al. (2021). We use temperature reconstructions representing annual or summer surface conditions (fig. 2).

Marine proxies generally display more heterogeneous LIG-PI changes than model outputs. In summer, the IPSL-CM6A-LR model simulates the surface warming well, but it not reproduces local cooling in the Labrador and Norwegian Seas. This mismatch appears to be a general feature across CMIP5 (Lunt et al., 2013; Masson-Delmotte et al., 2013) and CMIP6 (Otto-Bliesner et al., 2021) models. In previous studies, this has been attributed to uncertainties or simplifications in the specified boundary conditions. Indeed the PMIP4-CMIP6 protocol consists in setting the ice sheets to their modern configuration and

neglect the freshwater inputs to the North Atlantic from ice melting in the LIG simulation. These have been shown to be responsible for local heterogeneities in simulations of the Last Interglacial climate (Govin et al., 2012; Stone et al., 2016) : these freshwater fluxes modulate the strength of the Atlantic Meridional Overturning Circulation (AMOC), and thus, the inflow of warm Atlantic waters in the Arctic Ocean. Moreover, comparison with sedimentary data suggests that the IPSL-CM6A-LR model simulates too much sea ice in the Labrador Sea (Kageyama et al., 2021). With a larger sea ice cover in this region, air-sea

heat exchanges are reduced, which also influences the AMOC intensity (Pedersen et al., 2017; Kessler et al., 2020).

  The IPSL-CM6A-LR model and terrestrial data generally agree on the sign of the near-surface temperature anomaly.They often differ on its magnitude, but the amplitude of the reconstructed temperature anomalies is not always consistent for sites close to each other (at the scale of the model's saptial resolution), as it is the case in the North Atlantic Ocean in summer. The model does not capture the strong annual warming recorded in the NEEM ice core (NEEM community members, 2013). This bias

has already been identified in the evaluation of the IPSL-CM6A-LR present day climate. It could be slightly amplified by the prescription of a modern Greenland ice sheet in the LIG simulation.

  Despite model-data disagreements, the IPSL-CM6A-LR model and most of the models considered in Otto-Bliesner et al. (2021) converge towards a similar behaviour. By analysing the reasons for Arctic climate change in our simulations, our aim is also to contribute to understand the mechanisms of these climatic changes and how their representation could be improved



to obtain better agreement with the reconstructions. For this study, we consider that the model-data agreement is sufficient to investigate the processes contributing to the Last Interglacial Arctic warming.

## 2.3 Arctic energy budget framework

The energy budget framework has been developed to identify key dynamical processes contributing to the Arctic amplification from observations and reanalyses (Nakamura and Oort, 1988; Semmler et al., 2005; Mayer et al., 2017, 2019; Serreze et al.,

2019) or climate models (Rugenstein et al., 2013). We estimate the coupled atmosphere-ocean-land-sea ice energy budget of the Arctic during the LIG and the PI periods based on the works of Mayer et al. (2019) and Serreze et al. (2019),.

The seasonal cycle of this energy budget computed from model outputs averaged over the last 200 years of the simulations. We quantify the heat transfers between the surface and the atmosphere, the oceanic and atmospheric heat transports, and the heat storage terms over the Arctic region defined as the area between 60 and 90°N. A schematic representation of the different

contributions involved in the energy budget is displayed in Figure 3. All terms are expressed in W m$^{-2}$.

We consider the energy content of an atmospheric column from the surface to the top ($AHC$). It can be expressed as the sum of the internal ($C_{pa}T$), kinetic ($E_k$), latent ($L_eq$) and potential ($\phi_s$) energies. The time derivative of the atmospheric heat content yields the atmospheric heat storage ($AHS$) which varies with the radiative flux at the top of the atmosphere ($F_{TOA}$), surface heat flux ($F_{SFC}$) and the heat transport ($AHT$).

$$AHC = \frac{1}{g} \int_0^{ps} (C_{pa}T + E_k + L_eq + \phi_s)dp \tag{1}$$

$$AHS = \frac{\partial}{\partial t} \frac{1}{g} \int_0^{ps} (C_{pa}T + E_k + L_eq + \phi_s)dp \tag{2}$$

$$AHS = F_{TOA} - F_{SFC} + AHT \tag{3}$$

where $g$ is the gravitational acceleration (9.81 m s$^{-2}$), $C_{pa}$ is the specific heat of the atmosphere at constant pressure (1005.7 J K$^{-1}$ kg$^{-1}$), $T$ is the temperature (in Kelvin), $E_k$ is the kinetic energy computed as $\frac{u^2+v^2}{2}$ (in m$^2$ s$^{-2}$), $L_e$ is the latent heat of

evaporation (2.501 × 10$^6$ J kg$^{-1}$), $q$ is the specific humidity (in kg kg$-1$, $\phi_s$ is the surface geopotential (in m$^2$ s$^{-2}$), $p$ is the pressure (in Pa) and $ps$ is the surface pressure (in Pa).

The surface flux ($F_{SFC}$) can be decomposed into downwelling shortwave radiation ($SW_{dn}SFC$), upwelling shortwave radiation ($SW_{up}SFC$), downwelling longwave radiation ($LW_{dn}SFC$), upwelling longwave radiation ($LW_{up}SFC$), latent heat flux (*flat*) and sensible heat flux (*fsens*).

$$F_{SFC} = SW_{dn}SFC - SW_{up}SFC + LW_{dn}SFC - LW_{up}SFC + flat + fsens \tag{4}$$

Finally, the energy flux at the top of the atmosphere ($F_{TOA}$) is equal to the difference between upwelling and downwelling radiative fluxes :

$$F_{TOA} = SW_{dn}TOA - SW_{up}TOA - LW_{up}TOA \tag{5}$$





As for the atmosphere, the energy content of the ocean ($OHC$) is integrated from the surface to the bottom of the oceanic
column.

$$OHC = \rho_w Cp_w \int_0^z T_{cons} dz \tag{6}$$

$$OHS = \rho_w Cp_w \frac{\partial}{\partial t} \int_0^z T_{cons} dz \tag{7}$$

$$OHS = F_{SFC} \times f_{oce} - F_{BOT} \times f_{ice} + OHT \tag{8}$$

where $f_{oce}$ is the ocean area fraction, $f_{ice}$ is the sea ice area fraction, $\rho_w$ is the sea water density (1035 kg m$^{-3}$), $Cp_w$ is the
specific heat of the ocean at constant pressure (J K$^{-1}$ kg$^{-1}$), $T_{cons}$ is the sea water conservative temperature (in Kelvin) and
$z$ is the depth (in m). We use the sea water conservative temperature because it better represents the oceanic heat content than
the sea water potential temperature (Intergovernmental Oceanographic Commission et al., 2010). In equations 2 and 7, we use
a monthly time step to derive the atmosphere and ocean heat content respectively.

The sea ice bottom heat flux ($F_{BOT}$) represents the heat exchanges between the ocean and sea ice. It is defined as the difference
between the ocean heat flux ($F_{OCE}$) and the conductive heat flux ($F_{COND}$) at the bottom of the sea ice cover.

$$F_{BOT} = F_{OCE} - F_{COND} \tag{9}$$

The sea ice heat content primarily results from the heat exchanges between ice with the atmosphere and the ocean. The heat
flux derived from the sea ice transport ($IHT$) from regions of ice formation to regions of ice melt is included in the calculation
of the sea ice heat storage ($IHS$).

$$IHS = F_{SFC} \times f_{ice} + F_{BOT} \times f_{ice} + IHT \tag{10}$$

For terrestrial regions, energy budget variations are only due to changes in $F_{SFC}$ over the continents. Lateral heat transport
divergences are small and can be ignored (Serreze et al., 2019), thus the storage term is equal to the air-land heat exchanges.
We choose the same sign convention for all fluxes ie. positive fluxes point downward. They act to warm the surface when they
are positive except for $F_{BOT}$, which cools the sea ice when positive.

Ocean, atmosphere and sea ice transports are computed as the residual of the surface heat fluxes, the bottom heat flux and
the heat storage term. From equations 3, 8 and 10, we can write:

$$AHT = AHS + F_{SFC} - F_{TOA} \tag{11}$$

$$OHT = OHS - F_{SFC} \times f_{oce} + F_{BOT} \tag{12}$$

$$IHT = IHS - F_{SFC} \times f_{ice} - F_{BOT} \tag{13}$$

These equations give coherent zonally averaged profiles for the PI simulation with a zero northward atmospheric and oceanic
heat transport at the North Pole (fig. 4). To validate this approach, we also compared the mean annual OHT computed as





a residual with the OHT comuted online by the model (AHT and IHT are not are not stored in the CMIP6 database). The difference between both methods is around 0.005 PW for the Arctic region, which represents less than 2% of the model average.

The annual value of storage terms should be zero in the ideal case of an equilibrium climate. This is not the case for both simulations. The PI AHS and OHS are lower than the current observed energy imbalance of 0.5 W m$^{-2}$ in terms of absolute value (Roemmich et al., 2015; Hobbs et al., 2016). However, the LIG AHS and more specifically the LIG OHS are far above this reference value since they are respectively equal to 0.5 and 1.1 W m$^{-2}$. This "energy excess" may arise from assumptions made for the energy budget computation or from an ocean drift in the LIG simulation. The deep ocean does not seem to have

quite reached equilibrium in the LIG simulation. As illustrated by figure 5, the sea surface temperature and ocean heat content drifts are small, but more pronounced in the latter case. We can therefore assume that deep ocean temperature variations are not negligible.

We also estimate the energy provided by snowfall ($E_{SF}$) as defined by Mayer et al. (2017, 2019):

$$E_{SF} = L_f(T_p)P_{snow} \tag{14}$$

where $L_f(T_p)$ is the latent heat of fusion (–0.3337 × 10$^6$ J kg$^{-1}$) and $P_{snow}$ is the snowfall rate (in kg m$^{-2}$ s$^{-1}$).
We obtain an annual snowfall contribution to the atmospheric heat budget of 2.95 W m$^{-2}$ for the PI period and of 2.66 W m$^{-2}$ for the LIG period. The LIG-PI anomaly is very small compared to the anomaly of the other fluxes. Thus, we have decided to neglect the snowfall contribution in the rest of this study.

## 3    Results

In this section, we present anomalies defined as the difference between Last Interglacial (LIG) and pre-industrial (PI) simulated climatic fields.

### 3.1    Seasonal variations of the Arctic climate during the Last Interglacial period

The change in insolation between the LIG and the PI periods leads to a global mean annual anomaly of –0.19 °C. This value is

in the range of the PMIP3 multi-model mean estimate of 0.0 ± 0.5°C (Masson-Delmotte et al. 2013) and close to the PMIP4 multi-model mean value of –0.02 ± 0.32°C (Otto-Bliesner et al., 2021). In the Arctic region, the temperature response to orbital forcing results in a mean annual warming of 1.8°C compared to PI.

The surface air temperature anomaly displays substantial seasonal (fig. 6) and spatial variations (fig. 7). During the LIG, winter (DJF) and spring (MAM) seasons are about 0.2°C colder compared to the PI period. In both cases, most of this cooling

takes place over continents (fig. 7a, b). Conversely, over the Arctic ocean, the surface air temperature anomaly is positive, especially in areas where sea ice concentration decreases (fig. 7a, b, e and f). This difference between land and ocean is explained by the larger effective heat capacity of the ocean resulting in a greater amount of energy absorbed and stored by





the ocean. However, it should be noted that even if the seasonal average of the surface air temperature anomalies is similar in winter and spring when averaged over the whole Arctic region, the magnitudes of the anomaly are locally higher in winter.

Summer (JJA) and autumn (SON) are warmer during the LIG than during the PI period over both the ocean and the continents. The behaviour of the climatic fields is very different for both seasons. While the maximum warming occurs in summer over continental areas, over the oceanic regions, the largest temperature anomalies are found in autumn. There are also differences in the magnitude of the summer and autumn warmings. The temperature anomaly is expected to be especially large in summer when the insolation anomaly is the largest. Indeed, it reaches +3.2°C on average, but the autumn value is even larger and

reaches +4.0°C (fig. 6 and fig. 7c, d).

Surface air temperature anomalies are also associated with variations of snow cover and Arctic sea ice. The strong warming occurring during summer and autumn results in a large retreat of the Arctic sea ice cover, that persists during the rest of the year south of Svalbard and in the Barents Sea (fig. 7e-h). On the other hand, the snow cover does not appear to respond to the temperature rise in summer (fig. 7i-l). The snow cover anomaly is generally very low because the PI snow cover is already

relatively small during summertime. However, in autumn, the snow cover is strongly negative. The cooling in DJF and MAM reduces the effects of the summer and autumn polar amplification. Despite the slight decrease in temperature in DJF and MAM, sea ice does not fully recover after its strong decline in summer and autumn seasons. As a result, compared to the PI simulation, the sea ice area decreases by $0.5 \times 10^6$ km$^2$ in DJF and by $0.3 \times 10^6$ km$^2$ in MAM.

Figure 6 displays a time lag of four months between the maximum of insolation (in June) and the surface air temperature (in

October) anomalies. This lag has already been observed in previous studies investigating the future polar amplification (Manabe and Stouffer, 1980; Rind, 1987; Holland and Bitz, 2003; Lu and Cai, 2009; Kumar et al., 2010)). It suggests the existence of processes limiting the summer warming and/or feedbacks inducing a strong warming in autumn despite the decrease in insolation anomaly. In the following, we investigate the origin of this time lag. To achieve this, we analyse the respective roles of the atmosphere, ocean, sea ice and continental surfaces in summer (section 3.2), and in autumn (section 3.3).

## 3.2   The Arctic summer warming

In summer, the positive anomaly of near-surface air temperature reaches 3.2 °C over the Arctic and it is associated with a large retreat of the sea ice area of 1.9 million km$^2$ (see section 3.1). As previously mentioned, this warming corresponds to a strong insolation anomaly in the high latitudes of the northern hemisphere. It affects the entire atmospheric column with a maximum air temperature anomaly between 600 and 300 hPa reaching more than 5 °C (fig. 8a). At the top of the atmosphere,

downwelling shortwave radiation (SWdnTOA) increases by more than 25 W m$^{-2}$ on average compared to PI (fig. 9). This energy excess is uniformly distributed between 60 °N and 90 °N. Since solar forcing is only a function of latitude, it is similar over land and ocean. However, only 25% of this energy excess is absorbed by the ocean, and 50% by the continents.

### 3.2.1   Over the ocean

Over the ocean, a large amount of anomaly in solar energy does not reach the surface and is absorbed or reflected by the

atmosphere. As aerosols are prescribed, this may be attributed to changes in the distribution or the characteristics of the cloud





cover. Over the Arctic region, low-level clouds dominate (Shupe and Intrieri, 2004; Kay et al., 2016). They are often composed of both supercooled liquid water and ice. This type of cloud have a strong radiative effect on shortwave radiation, notably through the variations of the liquid water path, a measure of the weight of the liquid water droplets in the atmosphere above a unit surface area on the Earth (AMS glossary). Figure 10a shows a small cloud cover anomaly over the Arctic ocean but the

liquid water path increases (fig. 10d) causing more reflection of solar radiation back to space (see section 1). As the LIG-PI liquid water path anomaly is very high (more than 2 g m$^{-2}$) over the Arctic ocean, the effects of cloud on incident shortwave radiations seems to be fundamental to explain the differential energy received over the ocean compared to the continents.

To better quantify the total impact of clouds on the Arctic shortwave budget, we compute the shortwave cloud radiative effect (SW CRE), defined as the difference in shortwave fluxes between an atmosphere with and without clouds. In both LIG and PI

simulations, the SW CRE is negative over ocean (not shown), implying a strong cooling effect of clouds on the Arctic climate. The LIG SW CRE absolute value is about 31% higher than the PI one on average (not shown). This leads to a negative SW CRE anomaly over ocean in summer (fig. 11a), which is consistent with the liquid water path anomaly (fig. 10d).

Despite a small incident solar radiation (SWdnSFC) anomaly over the ocean, surface heat flux anomalies are sufficiently high to impact the sea ice cover (fig. 7g). As sea ice declines, more oceanic surface is exposed and can interact with the atmosphere

reducing the reflective power of the surface. Due to this albedo effect, the SWupSFC anomaly is negative over the ocean (fig. 9) and more solar radiation is absorbed and stored in the upper layers of the ocean. This results in a warming of the oceanic surface and a large increase of the ocean heat storage (fig. 9). According to the Planck's law, LWupSFC increases as a function of $\sigma T^4$. Consequently, ocean emits more longwave radiation compared to PI but the total longwave radiation (LWdnSFC-LWupSFC) anomaly is positive (fig. 9) and strengthens the warming of the oceanic surface. Considering all the heat fluxes at the air-sea

interface, the ocean receives 14.9 W m$^{-2}$ more energy than during the pre-industrial period. Turbulent heat fluxes show small variations, with a negative anomaly of –1.4 W m$^{-2}$ on average.

The surface heat budget over the ocean confirms that the upper layer of the ocean warms up during the LIG. Unlike the atmospheric warming that affect the entire atmospheric column, the increase in ocean temperatures only appears in the upper 100 m of the ocean (fig. 8). In addition, ocean heat transport (OHT) shows a significantly anomalous heat convergence towards

the high latitudes of the northern hemisphere. With a positive anomaly of more than 5 W m$^{-2}$ (fig. 9), it represents an important source of heat. In the PI simulation, OHT is negative, which means that heat is advected outside the Arctic basin to balance surface forcing (not shown). It becomes positive in the LIG simulation as surface heat flux increases also over the ocean (eq. 13). It implies that ocean is affected by thermodynamic and/or dynamic changes.

Changes in heat transport, surface heat budget and, to a lesser extent, sea ice-ocean heat flux contribute to increase the ocean

heat storage (OHS). The value of the oceanic storage nearly doubles compared to PI. This strong increase suggests that ocean is a key factor in the warming of the Arctic region in summer.

### 3.2.2 Over the continents

Over the continents, the positive SWdnSFC anomaly contributes to warm the surface and the lower atmosphere. This warming is amplified by a reduced negative shortwave CRE relative to the pre-industrial period (fig. 11a). It is caused by a decrease in



clouds cover and liquid water path over the continents relative to PI (fig. 10a and fig 10d). Changes in cloud characteristics have an adverse effect on incident SW radiations over the continents and ocean, and thus, they contribute to increase the land-ocean contrast in the near-surface air temperature anomaly.

The energy received at the surface is partly emitted back to the atmosphere through upwelling shortwave radiations (SWupSFC), upwelling longwave radiations (LWupSFC) and turbulent fluxes. The SWupSFC anomaly is very small relative to the other upward heat flux anomalies (fig. 14), because of small changes in surface albedo associated with slight variations in snow cover in summer. Due to the large SWdnSFC anomaly over the continents, the temperature rises significantly compared to PI leading to an LWupSFC anomaly of 12.8 W m$^{-2}$ (fig. 9) partially compensated by the positive LWdnSFC anomaly (9.3 W m$^{-2}$). As the anomaly of the longwave CRE (fig. 11b) is very weak, increase in LWdnSFC is not related to changes in cloud cover. However, it could be caused by increasing specific humidity in the atmosphere (fig 12). A greater amount of water vapor leads to a larger absorption of longwave radiations, which amplifies the greenhouse effect and then, the temperature.

Latent and sensible heat fluxes both contribute to the turbulent heat flux anomaly. Their respective contribution differs from one region to the other. Over Alaska, northeastern Canada, Siberia and Scandinavia, the latent heat flux anomaly is significant. It is not driven by snow sublimation as snow cover anomaly is very low, except in the Canadian archipelago (fig. 7c). However, the latent heat flux anomaly is correlated with the evaporation anomaly as shown in figures 10b and 10e. Where latent heat flux anomaly is negative or approaches 0 W m$^{-2}$, there is an enhancement of the sensible heat flux. Turbulence is generated in the boundary layer as wind speed intensifies over land surface (fig. 10.c and fig. 10.f). However, changes in surface wind speed do not appear to amplify the latent heat flux. An explanation of this could be that, in regions with a strong positive anomaly of surface wind speed and a negative anomaly of latent heat flux, there is less water in the soil to evaporate from the start.

The land energy budget confirms that the continental surfaces lose energy to the benefit of the atmosphere. The shortwave radiation anomaly warms the continents, which in turn transfer the energy back to the atmosphere through longwave radiation (3.5 W m$^{-2}$) and turbulent heat fluxes (8.7 W m$^{-2}$).

The atmospheric heat storage (AHS) increases by 7.1 W m$^{-2}$ compared to PI. AHS depends on changes in the internal, latent, kinetic and potential energy storage anomalies (eq. 2). Because of enhanced heat fluxes towards the atmosphere (SWdnTOA, LWupSFC and turbulent fluxes), the internal energy storage and to a lesser extent the latent and kinetic energy storage increase relative to PI leading to a higher atmospheric energy storage during the LIG of 7.1 W m$^{-2}$ (fig. 9). The AHS anomaly is less than the OHS anomaly mainly because of the much higher heat capacity of the ocean. Moreover, the atmospheric heat transport (AHT) does not contribute to the summer warming. Since it decreases from PI to LIG, the AHT anomaly almost compensates the OHT anomaly, as theorized by Bjerknes (see Swingedouw et al. (2009)). The AHT is partly reduced due to the decrease of the northern hemisphere meridional temperature gradient, and thus the decrease of the poleward dry static energy transport.

In conclusion, the ocean and the continents respond in different ways to the orbital forcing in summer. While the oceanic surface tends to warm up as it better absorbs solar radiation, the continental surface provides energy back to the atmosphere. LIG summer warming is directly due to orbital forcing changes and to heat exchanges between the atmosphere and the continents surrounding the Arctic ocean. Processes detailed in this section for the Arctic summer warming are summarized in figure 13.





### 3.3 The Arctic autumn warming

Despite a small insolation anomaly (fig. 1), the strongest surface warming occurs in autumn (fig. 7a). Figure 8 shows that the warming does not extend over the entire atmospheric column as in summer. It is confined in the lower layers of the atmosphere below 800 hPa.

In autumn, the LIG insolation is similar to the PI one (fig. 1). As a consequence, the shortwave radiation anomalies (SWdnTOA, SWdnSFC and SWupSFC) do not contribute much to the total energy budget anomaly compared to summer (fig. 14 compared to fig. 9). By contrast, longwave radiation anomalies play a crucial role in the autumn warming. Larger longwave fluxes are also emitted into the atmosphere because open ocean waters are warmer than the cold sea-ice surface. The LWupSFC anomaly is 11 W m$^{-2}$ on average (fig. 14) and peaks at more than 40 W m$^{-2}$ over the East Siberian and the Kara Seas (fig. 15d). Similarly to the summer months, the LWdnSFC anomaly is stronger than the LWupSFC anomaly resulting in a positive longwave radiative budget. The increase in specific humidity over the ocean (fig. 12) is likely related to the large retreat of the sea ice cover and associated evaporation (fig. 7h and fig. 15f). In response to increasing humidity in the atmosphere (fig. 12), the Arctic cloud cover expands (fig. 15c) leading to a positive cloud feedback over the Arctic ocean (fig. 11b). This longwave cloud radiative effect (LW CRE, computed in a similar way that the SW CRE) favours the autumn warming by trapping outgoing LW radiations in the atmosphere (Schweiger et al., 2008; Goosse et al., 2018).

The surface air temperature anomaly and the additional heat absorbed by the upper ocean during summer amplify the retreat of the sea ice edge. The autumn months experience the largest sea ice decline with a sea ice area anomaly of $-3.5 \times 10^6$ km$^2$ (fig. 7h). This reveals large open water areas which favour heat transfers from ocean to the atmosphere. Figure 14 indicates that turbulent heat fluxes slightly increase (by 1.9 W m$^{-2}$ on average over the ocean). However, at the local scale their contribution is larger (fig. 15c and 15f). In regions where the sea ice loss is the greatest (ie. along the Siberian and Alaskan coasts, and over the Barents and Greenland Seas), the sum of sensible and latent heat fluxes reaches more than 20 W m$^{-2}$ (fig. 7h). Over the continental areas, the turbulent heat fluxes anomaly seems to have no significant impact on the surface heat budget (fig. 14).

Despite the strong Arctic warming, the atmospheric energy storage (AHS) anomaly is negative meaning that the atmosphere looses more energy than for the PI period. During the autumn, the internal energy storage ($-4.9$ W m$^{-2}$) and the potential energy storage anomalies ($-4.5$ W m$^{-2}$) contribute significantly to the energy loss (table 2). The anomaly of the internal energy storage depends on air temperature fluctuations from one season to the other. As illustrated in figure 8, the air temperature increases from summer to autumn near the surface but over the rest of the atmospheric column, the air temperature peaks in August. The potential energy storage anomaly is also strongly dependent on the temperature in the atmospheric column and follows the same trend as the internal energy storage.

Moreover, poleward oceanic and atmospheric heat transports weaken (fig. 14). This modulates the warming of the northern hemisphere high latitudes and does not contribute to the observed temperature increase.

Processes of the Arctic autumn warming are summarized in figure 16.

In summary, the Arctic region continues to experience the effects of the preceding summer warming through ocean and sea ice feedbacks during the autumn. Sea ice cover changes allow the ocean to release heat leading to a significant warming of the





surface atmospheric layer. As illustrated by figure 8, feedbacks operate in the lower atmosphere but not above. Yin and Berger

(2012) first explained such process using a surface heat budget analysis with the LOVECLIM-LLN model, which they called the "summer remnant effect". This "summer remnant effect" modifies the seasonal impact of the astronomical forcing. In the IPSL-CM6A-LR model, it appears during the autumn and continue until winter to a lesser extent.

## 3.4    The Arctic sea ice mass variations

Sections 3.2 and 3.3 highlight the key role of the Arctic sea ice on the ocean and atmosphere heat balance. In particular, the

expansion of the sea ice cover determines the amplitude of the air-sea exchanges through variations of the surface albedo and sea ice insulating effect. During the LIG, the sea ice cover is reduced all year round relative to the PI period, reaching a peak of $-3.4 \times 10^6$ km$^2$ in autumn (fig. 6 and fig. 7e-h). The sea ice mass decreases too and lose about 0.08 Gt on annual average. To better understand the causes of this decline, we compute the mass budget terms anomalies using new diagnostics developed for the lastest Coupled Model Intercomparison Project CMIP6 (Notz et al., 2016; Keen et al., 2021). The mass budget terms

analysed here are the following: the basal growth, the ice formation in supercooled open water (or frazil), the melting at the top surface of the ice, the melting at the base of the ice, ice formation due to the transformation of snow to sea ice, the change in ice mass due to evaporation and sublimation and the ice advection into or out of the Arctic domain. The IPSL-CM6A-LR model outputs do not contain explicit lateral melt. These different processes are represented in figure 17 on summer and autumn for both PI and LIG periods. In summer, the main process by which sea ice is lost is basal melt during both periods. However, the

LIG-PI surface melting anomaly is higher than the LIG-PI basal melting anomaly. Thus, in summer, changes in sea ice volume are mainly related to changes in incident shortwave radiations rather than to changes in ocean-sea ice energy exchanges. In autumn, ice melt and growth processes are less strong during the LIG. The large autumn sea ice retreat (fig. 7h) is therefore not caused by increasing melt. It is the consequence of the substancial loss of sea ice during the previous summer which is exacerbated by the poor recovery of the sea ice cover in autumn. On figure 17, we can also note that sea ice advection is

surprisingly low, while Keen et al. (2021) show that it is one of the main factor of the mass budget variations.

## 4    Discussion

As seen before, polar clouds greatly influence the cooling or the warming of the atmosphere. Despite their importance in the global energy budget, climate models have difficulties to represent the coexistence of the two phases and often underestimate the supercooled liquid water proportion. Regarding the IPSL model, improvements in shallow convective scheme and

phase-partitioning in mixed phase clouds between LMDZ5A and LMDZ6A lead to an increase of supercooled droplets and a better distribution between low-level and mid-high clouds, which is more consistent with the most recent satellite observations (Madeleine et al., 2020). These improvements as well as a refined model tuning (Hourdin et al., 2020) result in a reduction of shortwave and longwave cloud radiative effects (CRE) in the mid to high latitude regions, which are in good agreement with the observations. However, while the distribution of liquid droplets and ice crystals in cold mixed phase clouds is closer to

observations, low-level clouds remain too abundant over high-latitude regions. The increase in low-level clouds seems to be





compensated by the decrease in the mid to high-level clouds, and finally does not impact the LW CRE.

On the other hand, the temperature biases described in section 2.2 can largely impact the surface heat budget, either directly through biases in longwave fluxes emitted by the Earth's surface or indirectly through changes in sea ice. The warm bias over the Arctic ocean favours the retreat of the sea ice edge, which is highly sensitive to surface temperature changes. In their

evaluation of the IPSL-CM6A-LR model, Boucher et al. (2020) compare sea ice area and extent in the *historical* simulations (1850-2014) with recent satellite observations. For both summer and winter, simulated results are slightly underestimated compared to satellite data, but are still within observational uncertainty. During the LIG, this bias only subsists in summer especially in the northernmost areas (Kageyama et al., 2021). In winter, model-data are in better agreement except for two sites in the Labrador Sea. At these locations, the model shows seasonal or perennial sea ice, while cores show ice-free conditions all

year round (Kageyama et al., 2021).

On the basis of a simple linear regression model, we try to identify the relationship between surface temperature biases in the *historical* simulation and those in the *lig127k* simulation. The aim is to determine if the model biases found in Boucher et al. (2020) are correlated with those of the *lig127k* simulation. Surface temperature biases are analysed at the core site location (fig. 2). As in Boucher et al. (2020), near-surface air temperatures simulated by the *historical* simulation are compared with

ERA-INTERIM dataset for the period 1980-2009 (Dee et al., 2011) and sea surface temperature with WOA13-v2 dataset for the period 1975–2004 (Locarnini et al., 2013). For the annual average, the lack of data points limits the interpretation of the linear regression and we cannot conclude on the impact of the model biases on the *lig127k* simulation (fig. 18). For the summer average, the correlation coefficient is low ($r^2$=0.18), which indicates that the model biases has only a limited influence on the *lig127k*. This result depends largely on the uncertainties on the reconstructions, which can be very large for some points. The

uncertainty associated to the surface temperature biases during the Last Interglacial is plotted on figure 2. It is estimated from the data uncertainty ($\pm 1\ \sigma$) and the standard deviation of the model outputs computed following a Gaussian distribution. Even though the correlation coefficient is low, figure 18b shows that the signs of the biases for modern day and LIG are generally consistent: there are only two sites for which the present bias is positive while the LIG bias is clearly negative, taking the uncertainties on the LIG reconstructions into account. This remarkable fact calls for further investigation, in a forthcoming

study.

## 5 Conclusions and Perspectives

In this work, we present an analysis of the seasonal cycle of the Arctic energy budget during the Last Interglacial period using IPSL-CM6A-LR model outputs.

In autumn, the near-surface air temperature anomalies are higher than in summer: there is a time lag between the maximum

anomaly of temperature (October) and the maximum anomaly of insolation (June). The summer warming is directly linked to the insolation anomaly and thus, to the anomaly in shortwave radiations received at the surface. Surface air temperature anomaly is higher over the continents which receive more solar radiations and release more heat back to the atmosphere through longwave radiation and turbulent fluxes. This warming persists in autumn as a result of different feedbacks involving



the ocean and sea ice. The Arctic Ocean and marginal seas, which play the role of a heat sink in summer, release heat back
to the atmosphere. This effect is amplified by the sea ice edge retreat and by the water vapor feedback. Anomalies of sea ice
cover and sea ice mass are negative throughout the year. The maximum ice loss is observed in the marginal seas in autumn. It
is the result of increasing basal melt in summer and decreasing basal growth in autumn compared to PI.

Our simulations do not account for climate–vegetation feedbacks nor for climate–ice-sheet feedbacks, as vegetation and ice-
sheet are prescribed in IPSL-CM6A-LR *pi-Control* and *lig127k* simulations. Changing land cover would modify both the
shortwave and longwave radiative budget. Pollen and macrofossil evidences for the LIG indicate boreal forests extending
northward and replacing Arctic tundra (CAPE Members, 2006; Schurgers et al., 2007; Swann et al., 2010). The expansion of
trees cause a decrease in surface albedo by partly masking snow and an enhancement of water vapour release to the atmosphere
through evapotranspiration. Therefore, additional simulation would be necessary to quantify the vegetation feedback on the
energy budget. On the other hand, variations of the LIG Greenland ice sheet geometry compared to PI one could also modify
the radiative budget through the albedo and elevation feedbacks. Moreover, they could also alter the sea surface conditions
and the oceanic circulation through freshwater release. Govin et al. (2012), Capron et al. (2014) and Stone et al. (2016) have
shown that freshwater inputs to the North Atlantic from the Greenland ice sheet mass loss improve model simulations with
respect to sediment and ice core data. However, accounting for the evolution of the Greenland ice sheet was not included in the
PMIP4-CMIP6 protocol of the *lig127k* simulation followed here.

Investigating the energy budget of other PMIP4-CMIP6 *lig127k* simulations would allow to evaluate whether their temperature
response to *lig127k* forcings is related to the same processes in terms of energy budget, and to compare the strengths of these
processes, especially in models which simulate a near complete loss of Arctic sea ice in summer (Kageyama et al., 2021).
Finally, a next step will be to put the dynamical processes highlighted in this study into perspectives by comparing them with
the processes of the Arctic future warming.

*Data availability.*   The original output data from the model simulations used in this study are available from the Earth System Grid Federation
(https://esgf-node.llnl.gov/, last access: June 11, 2021).

*Author contributions.*   MS, MK and SC designed the study. PB performed the *lig127k* simulation. MS produced all model figures and wrote
the papers under supervision of MK and SC. JBM contributed to the analysis of changes in cloud properties. All authors read the manuscript
and commented on the text.

*Competing interests.*   The authors declare that they have no conflict of interest.



*Acknowledgements.* This work was granted access to the HPC resources of IDRIS under the allocation 2016-A0030107732, 2017-R0040110492, and 2018-R0040110492 (project gencmip6) made by GENCI (Grand Equipment National de Calcul Intensif). It also benefited from the ES-PRI (Ensemble de Services Pour la Recherche à l'IPSL) computing and data centre (https://mesocentre.ipsl.fr, last access: June 11, 2021), which is supported by CNRS, Sorbonne Université, Ecole Polytechnique and CNES and through national and international grants.

MS is funded by a scholarship from the *Commissariat à l'énergie atomatique et aux énergies alternatives* (CEA) and the *Convention des Services Climatiques* from IPSL. MK is supported by the *Centre national de la recherche scientifique* (CNRS). SC and PB are supported by the CEA. JBM is supported by *Sorbonne Université* (SU).



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

710





| Astronomical parameters | LIG | PI |
|---|---|---|
| Eccentricity | 0.039378 | 0.016764 |
| Obliquity | 24.040° | 23.459° |
| Perihelion-180° | 275.41° | 100.33° |
| Date of vernal equinox | March 21 at noon | March 21 at noon |
| **Trace gases** | | |
| CO2 | 275 ppm | 284.3 ppm |
| CH4 | 685 ppb | 808.2 ppb |
| N2O | 255 ppb | 273 ppb |

**Table 1.** Astronomical parameters and atmospheric trace gas concentrations used to force LIG and PI simulations. From Otto-Bliesner et al. (2017).

| | DJF | MAM | JJA | SON |
|---|---|---|---|---|
| Atmosphric heat storage | −1.8 | 8.6 | 7.2 | −11.1 |
| Internal energy storage ($\frac{\delta}{\delta t}\int_0^{ps} Cp_aTdp$) | −1.2 | 3.9 | 3.9 | −4.9 |
| Latent energy storage ($\frac{\delta}{\delta t}\int_0^{ps} L_eqdp$) | −0.2 | 0.7 | 1.7 | −1.6 |
| Kinetic energy storage ($\frac{\delta}{\delta t}\int_0^{ps} E_kdp$) | $\sim 0$ | $\sim 0$ | $\sim 0$ | $\sim 0$ |
| Potential energy storage ($\frac{\delta}{\delta t}\int_0^{ps} \phi_sdp$) | −0.3 | 4.0 | 1.5 | −4.6 |

**Table 2.** Seasonal anomalies of the atmospheric heat storage and its components (W m$^{-2}$).





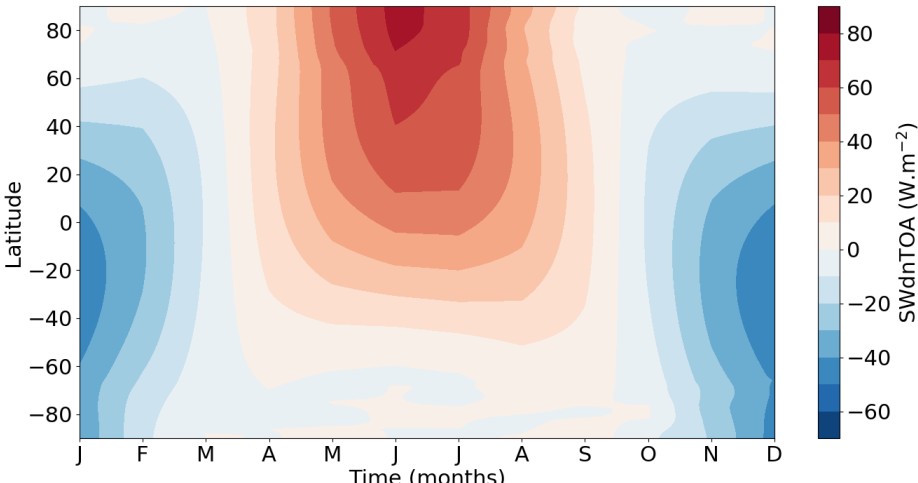

**Figure 1.** Annual cycle of the insolation anomaly (W m$^{-2}$) as a function of latitudes. The LIG insolation is computed using the celestial calendar with vernal equinox on March 21$^{th}$ at noon.

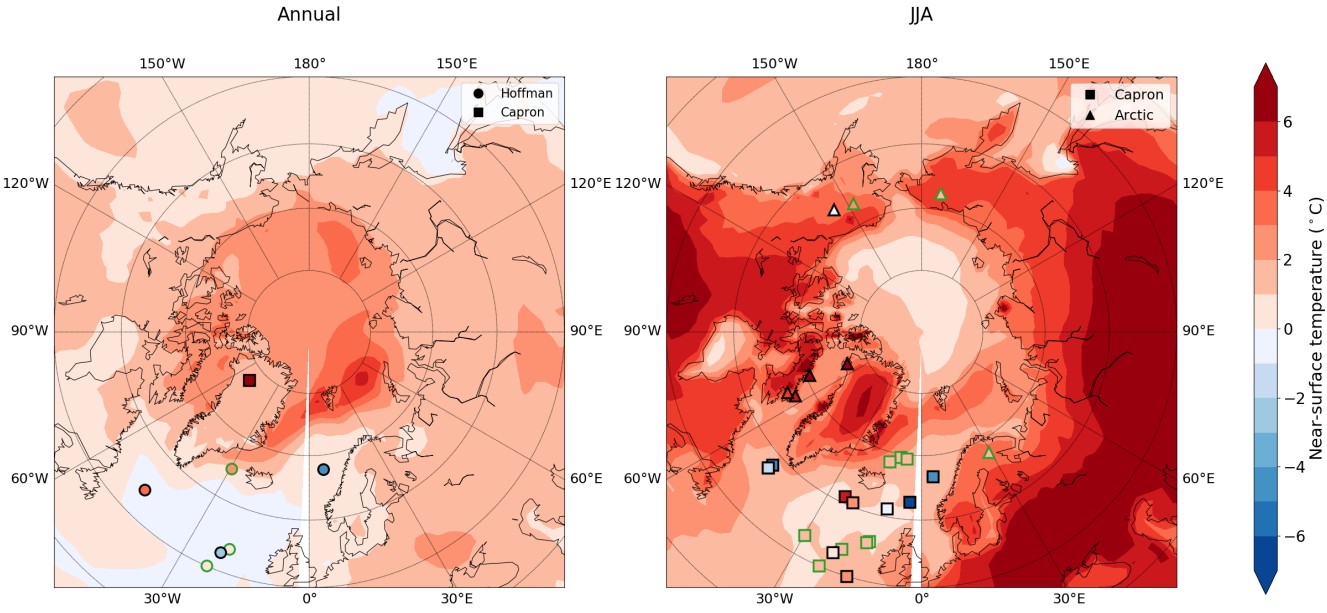

**Figure 2.** Annual (right) and summer (left) surface air temperature comparison (°C) between reconstructions and the IPSL-CM6A-LR outputs above 50°N. The green contour indicates a good agreement between data proxy and model, considering a data uncertainty of $\pm 1\sigma$. This figure is plotted from the synthesis provided by Otto-Bliesner et al. (2021). Markers represent the source of surface air reconstruction : circles for the compilation by Hoffman et al. (2017), squares for the compilation by Capron et al. (2014, 2017) and triangles for the Arctic compilation.



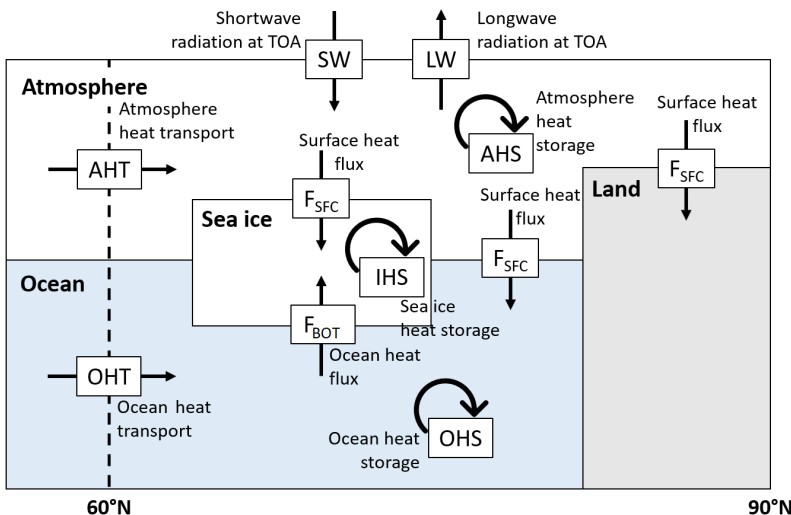

**Figure 3.** Representation of the different processes involved in the Arctic energy budget.

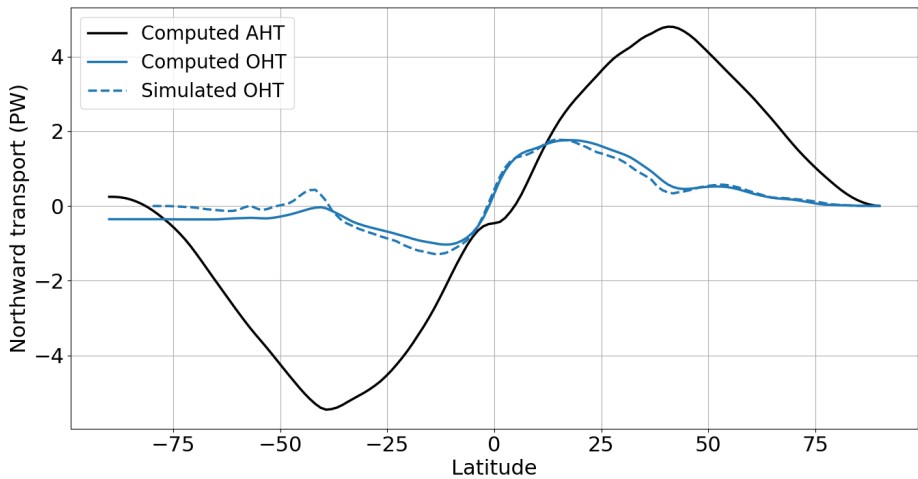

**Figure 4.** Annual zonal mean northward heat transport (PW). The northward heat transported computed as residal are represented by a solid line. The atmospheric heat transport is in black and the oceanic transport is in blue. The oceanic heat transport simulated by the IPSL-CM6A-LR model is represented by the dashed blue line.





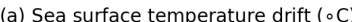

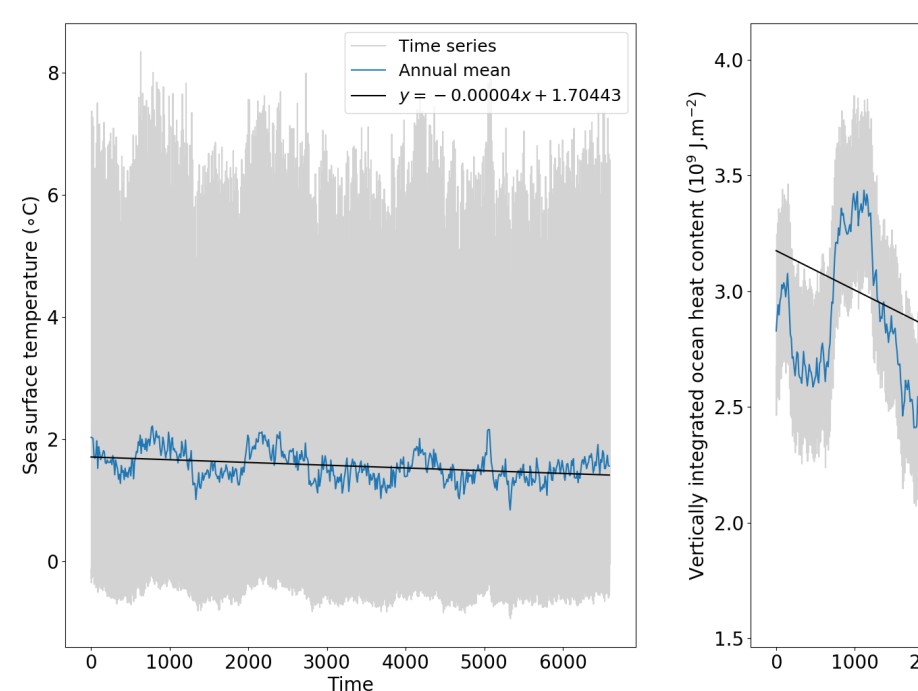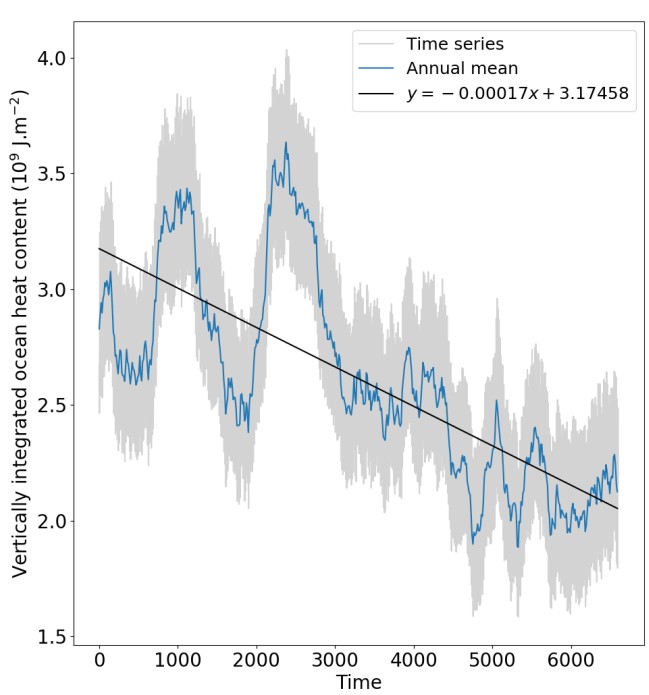

**Figure 5.** Time series of (a) the sea surface temperature (°C) and (b) the vertically integrated ocean heat content (J m$^{-2}$) averaged over the Arctic region (60–90°N). The time axis indicate the number of months since the year 1850.

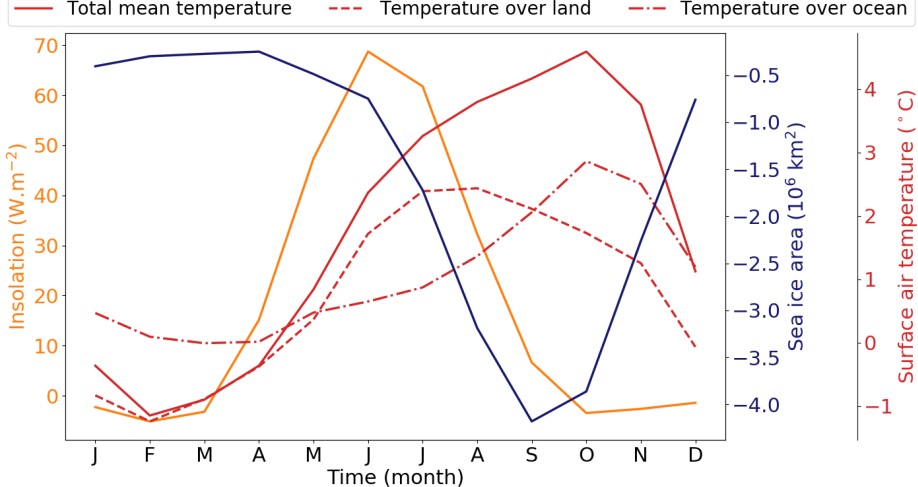

**Figure 6.** Annual cycles of solar radiation (orange line), surface air temperature (red lines) and sea ice area (dark blue line) anomalies. Variables are averaged between 60°N and 90°N





**Figure 7.** Seasonal cycles of the near-surface air temperature (a-d), sea ice concentration (e-h) and snow cover (i-l) anomalies. The value mentioned above the maps is the spatial average of the three variables over the Arctic (60 °N–90 °N). Seasons are abbreviated by: DJF = December-January-February (winter); MAM = March-April-May (spring); JJA = June-July-August (summer); SON = September-October-November (autumn).

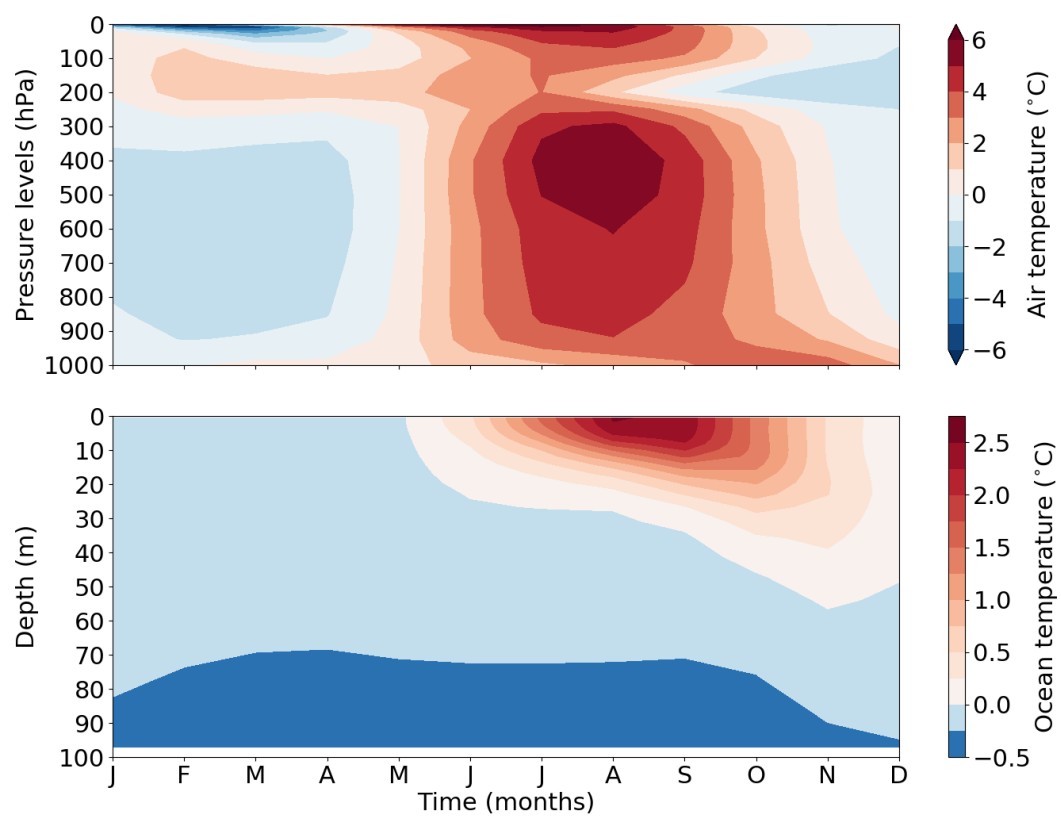

**Figure 8.** Annual evolution of temperature averaged over the Arctic (60 °N–90 °N), as a function of pressure in the atmosphere (top) and depth in the ocean (bottom). Below 100-m depth, the ocean temperature anomaly is negative ranging between 0 and -0.53 °C.



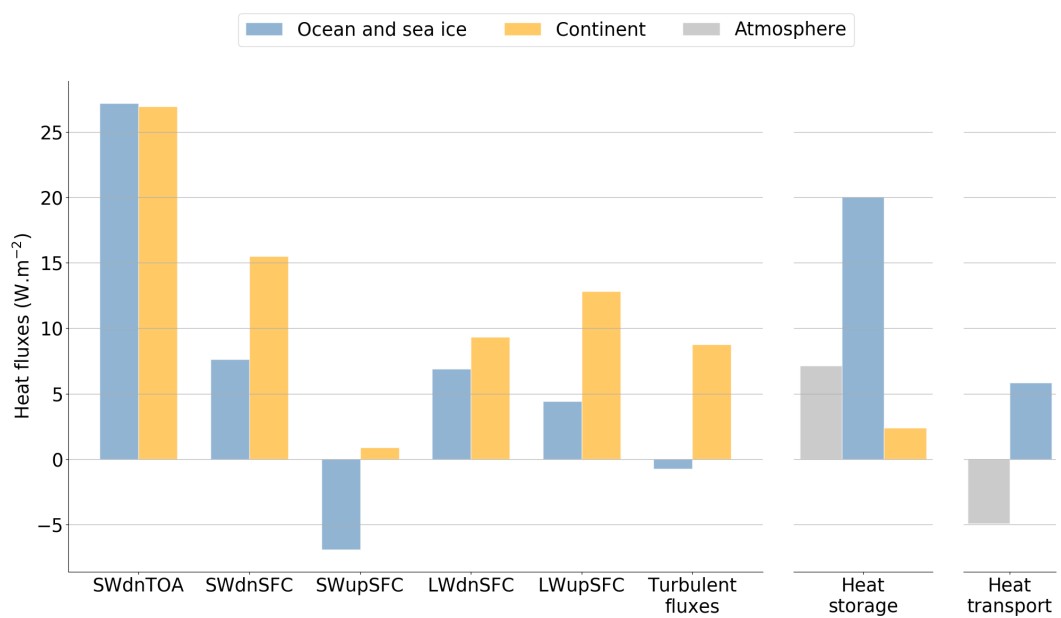

**Figure 9.** Quantification of the LIG-PI anomaly of surface heat fluxes (left), storage terms (center) and oceanic and atmospheric heat transport (right) for summer. Each flux included in the surface heat budget computation is plotted: solar radiation received at the surface (SWdnSFC), solar radiation reflected by the surface (SWupSFC), longwave radiation emitted by the surface (LWupSFC), longwave radiation emitted by the atmosphere (LWdnSFC) and turbulent fluxes given as the sum of latent and sensible heat flux. Variables are average between 60 °and 90°N. The surface heat fluxes anomalies are positive when the flux is stronger during the LIG.

**Figure 10.** Summer LIG-PI anomalies of (a) total cloud cover (%), (b) latent heat flux (W m$^{-2}$), (c) sensible heat flux (W m$^{-2}$), (d) liquid water path (kg m$^{-2}$), (e) evaporation (mm day$^{-1}$) and (f) surface wind speed (m s$^{-1}$)

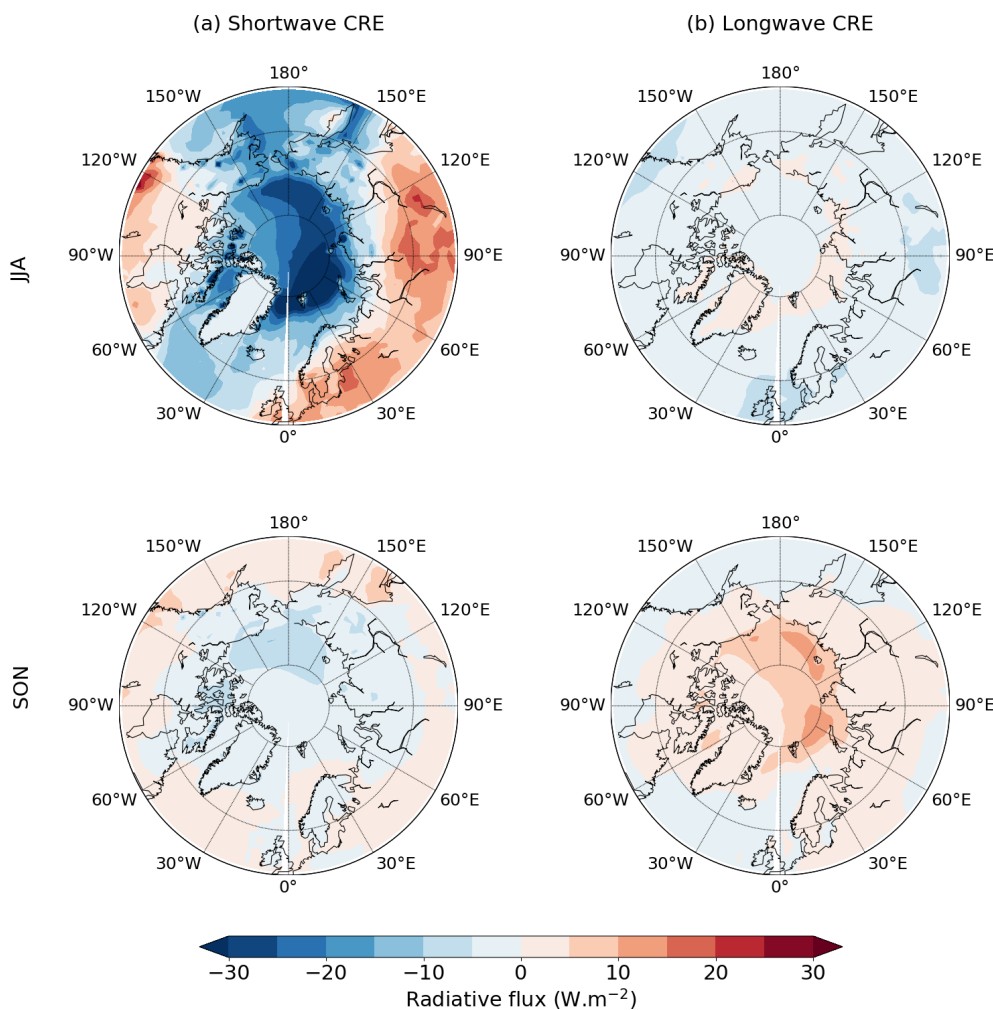

**Figure 11.** Summer (JJA) and autumn (SON) anomalies of (a) the shortwave cloud radiative effect (SW CRE) and (b) the longwave cloud radiative effect (LW CRE). All radiative fluxes are in W m−2.



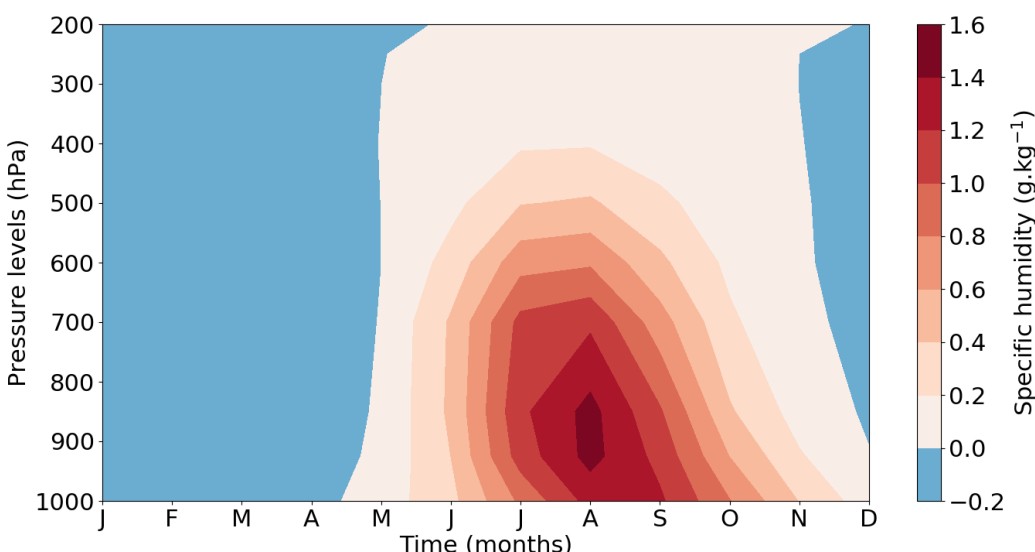

**Figure 12.** Annual evolution of the specific humidity anomaly (g kg$^{-1}$) according to pressure levels (hPa). The specific humidity anomaly is represented from the surface to 200 hPa and is spatially averaged over the Arctic region (60 °-90 °N).

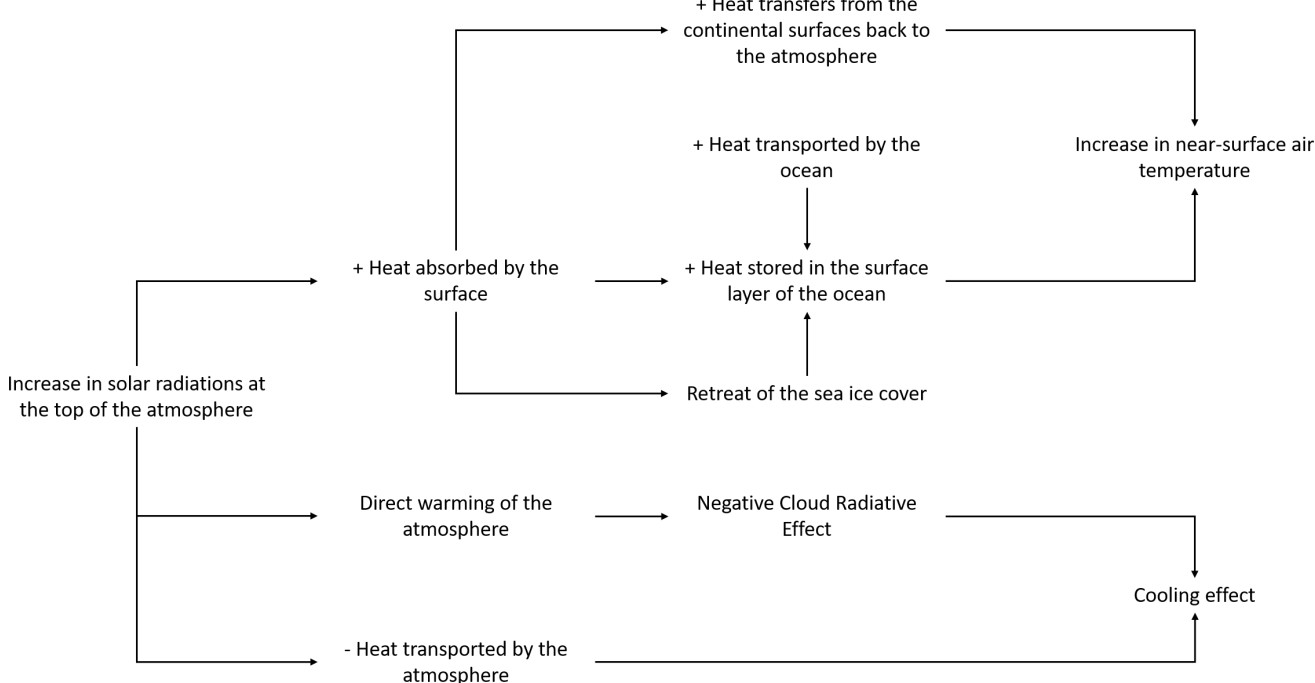

**Figure 13.** Diagram of climate processes and feedbacks involved in the Arctic summer warming.





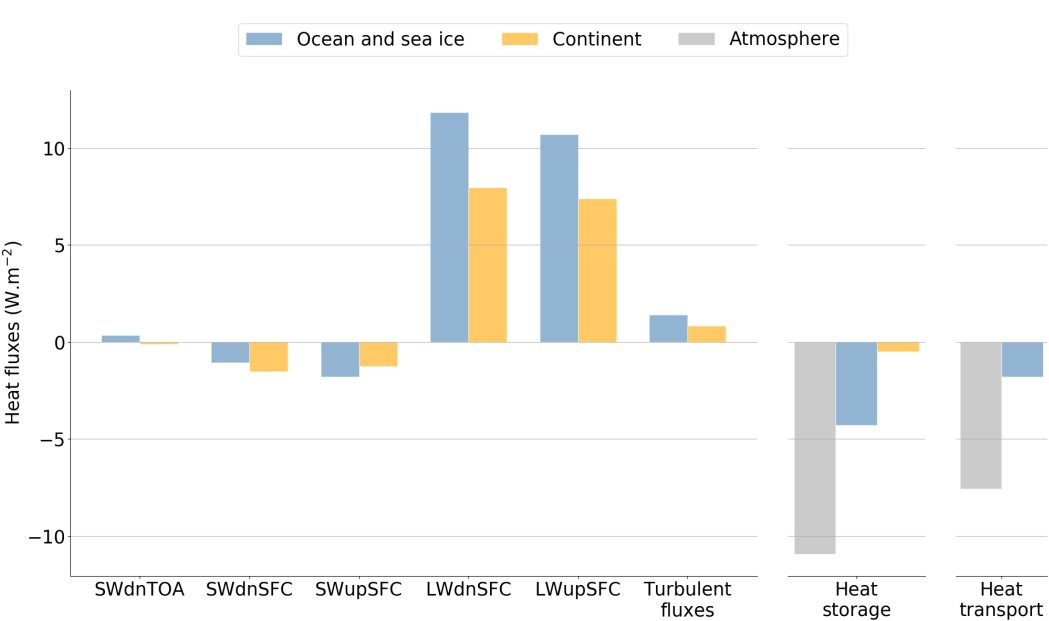

**Figure 14.** Same as figure 9 for SON.



**Figure 15.** Autumn LIG-PI anomalies of (a) longwave radiation emitted by the atmosphere, (b) latent heat flux, (c) totl cloud cover, (d) longwave radiation emitted by the surface, (e) sensible heat fluxes and (f) evaporation. All heat fluxes are in W m$^{-2}$, the total cloud cover in % and the evaporation in mm day$^{-1}$.





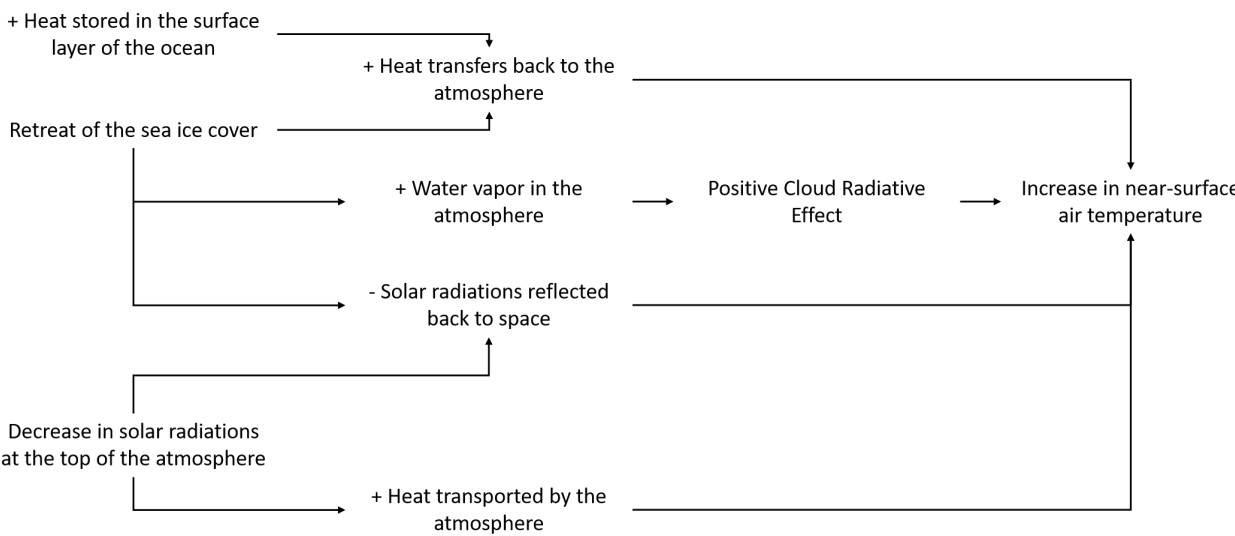

**Figure 16.** Diagram of climate processes and feedbacks involved in the Arctic autumn warming.

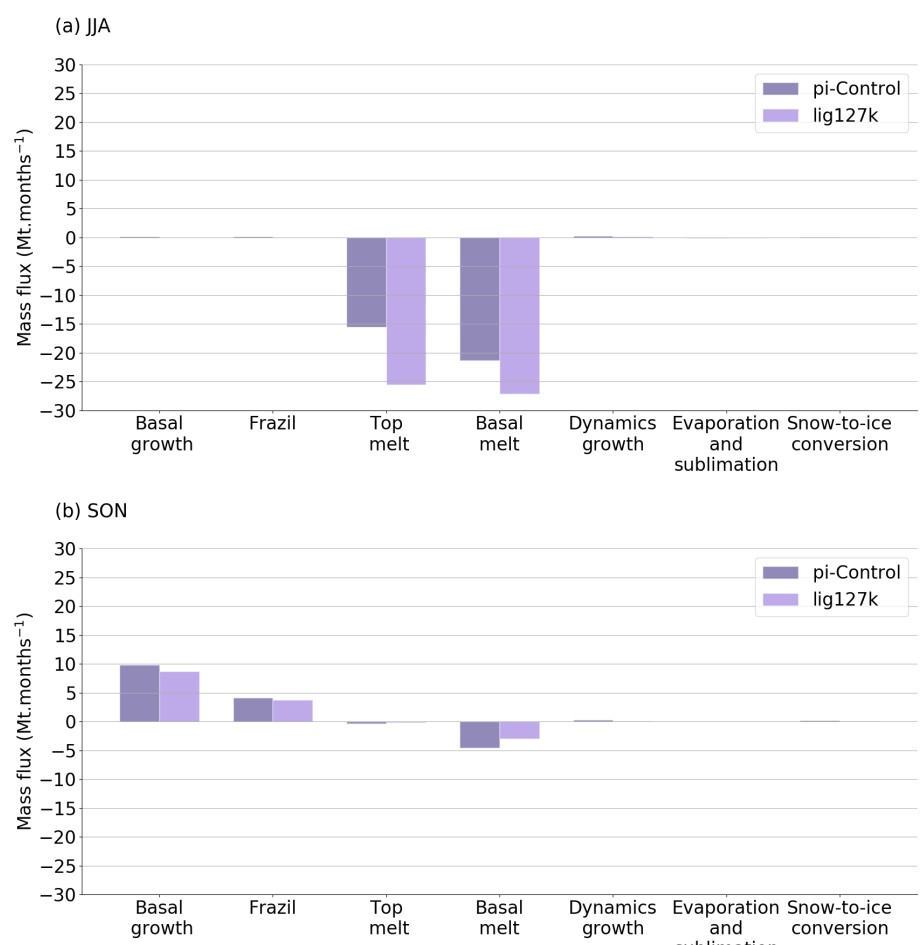

**Figure 17.** Components of the Arctic sea ice mass budget (Mt months$^{-1}$) in (a) JJA and (b) SON. They are computed for both the PI (dark purple) and LIG (pale purple) periods.





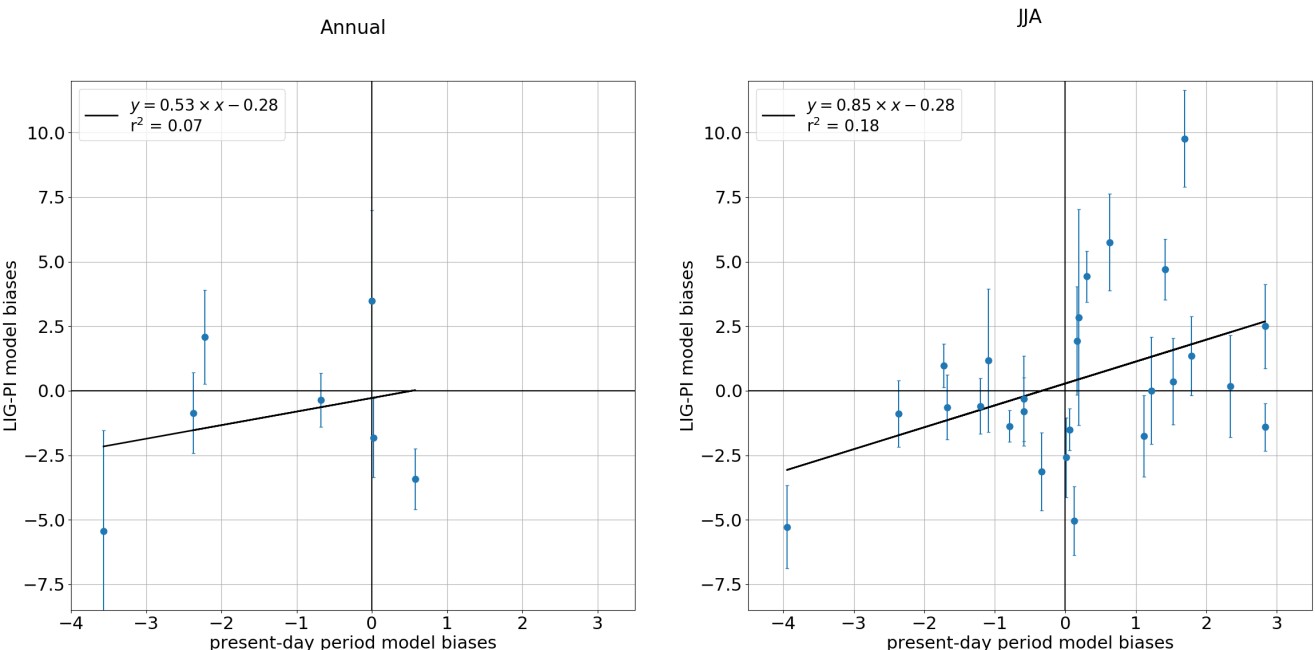

**Figure 18.** Linear regression of surface temperature biases in the *historical* simulation versus surface temperature biases in the *lig127k* simulation. Blue circle markers represent the model biases at LIG terrestrial and marine ice core sites. The coefficient of correlation ($r^2$) is calculated for each regression line. To compute model biases for the historical simulation, we use ERA-INTERIM near-surface air temperature data (1980–2005), the WOA13-v2 ocean temperature data (1985–2004) and the the first member of the IPSL-CM6A-LR *historical* simulations. The error bars are plotted in blue and represent the uncertainty on the surface temperature biases during the Last Interglacial.