# Peer review of "An energy budget approach to understand the Arctic warming during the Last Interglacial"

_Climate of the Past, 2021_

## Author Comment (AC1)

**Response to Pepijn Bakker**

We would like to thank the reviewer for his comments and suggestions that will guide our revisions and improve this paper. We hope that the revisions we plan to implement should satisfactorily address his comments.

Below, the reviewers' comments are highlighted in italic, and our responses follow in blue.

Marie Sicard on behalf of the co-authors
* * *
*The manuscript presents a thorough analysis of the changes in the Arctic climate simulated for the LIG. By doing so they manage to pinpoint the causes of the changes. The manuscript is well written and the authors have managed well to ensure that it reads well despite the lengthy and technical results. In the following I will list my comments and suggestions.*

*Main comments:*

*On lines 157-158 'similar behavior' is mentioned when it comes to simulating the LIG climate with the IPSL-CM6A-LR model or with the PMIP4 models presented by Otto-Bliesner et al. (2021). This is an important statement because it would imply that the results presented in this manuscript are more widely applicable to the PMIP4 LIG simulations. First of all, it should be made more clear in the manuscript what is meant with 'similar behavior'. Moreover, when looking at temperature and sea-ice changes for the PMIP4 ensemble in Otto-Bliesner et al. (2021), I see large differences. The magnitude of JJA and SON sea-ice anomalies varies widely and for winter (DJF) models even differ in the sign of the sea-ice anomalies. I think a short summary of the findings of Otto-Bliesner et al. (2021) related to sea-ice and the Arctic is needed in the current manuscript to put the results and analysis of the IPSL-CM6A-LR model in perspective.*

The reviewer is right. We agree that the term "similar behaviour" is confusing. Here, "similar behaviour" refers to the common feature shared by PMIP4 models to reproduce near-surface temperature anomaly suggested by the temperature synthesis provided by *Otto-Bliesner et al., 2021*. We propose to remove the following sentence:

"Despite model-data disagreements, the IPSL-CM6A-LR model and most of the models considered in *Otto-Bliesner et al. (2021)* converge towards a similar behaviour."

In order to put in perspective, the results of the IPSL-CM6A-LR model with those of other PMIP4 models, we propose to add these few lines at the end of section 4:

"*Otto-Bliesner et al. (2021)* have highlighted the large differences in the magnitude of high latitudes near-surface temperature anomalies among PMIP4 climate models that have run the lig127k simulation. On an annual scale, Arctic near-surface temperature changes range from -0.39 to 3.88°C. Models that simulate the most intense surface warming also show the largest reductions in minimum Arctic sea ice area (*Kageyama et al., 2021*; *Otto-Bliesner et al., 2021*). There is a large spread across models for the simulated summer Arctic sea ice area, with minimum sea ice area anomalies ranging from 0.22 to 7.47 x 10$^6$ km$^2$ for the Last Interglacial. Moreover, there is no consensus about the sign of winter sea ice area variations, with three models simulating a decrease in sea ice area during this season.

Furthermore, the effective climate sensitivity (ECS) of PMIP4 models varies from 1.8 to 5.6°C (Otto-Bliesner et al.,2021). The IPSL-CM6A-LR model is in the higher range with an ECS value of 4.6°C. However, this model does not simulate a strong annual Arctic warming and a large summer Arctic sea ice retreat compared to other models with high ECS values such as EC-Earth3-LR (4.2°C) or HadGEM3 (5.6°C)."

*Related to the comments above, I have a further comment on the role of clouds, the difficulty to model them and the 'robust' PMIP results for the LIG. A large role is determined here for the radiative effects of clouds. The role of clouds in the climate system is one of the things that are generally seen as very uncertain in models, so it seems surprising that PMIP3/4 models would show similar cloud effects for the Arctic in the LIG. How does one reconcile that? Does it imply that certain cloud feedbacks are in fact quite robust in models?*

Low-level clouds strongly impact the Arctic energy budget, primarily through their effects on incident solar radiation at the surface (SWdn). As shown in Kageyama et al. (2021), while the insolation received at the top of the atmosphere is similar for all models following the PMIP4 lig127k protocol, the amplitude of the SWdn annual cycle anomaly varies across PMIP4 models. The atmospheric energy budget could be analysed for only eight models (out of the initial 17 models), for which the data was available. Even with this reduced data set, the diversity of responses suggests that cloud feedbacks are not consistent in these climate models.

This information will be added to section 4.

**Minor comments:**

*As we know, the LIG is not a direct analogue for the future. Most importantly, insolation in winter was lower, quite different from ongoing and future CO2-driven warming. However, this study suggest that it is mostly the summer and autumn seasons that determine the differences in the Arctic climate compared to PI. Does that mean that in this specific context, the LIG does provide a rather good analogue? Can such a statement be made?*

From this study, it is difficult to conclude whether the Last Interglacial provides a rather good analogue for the future Arctic warming because we do not directly compare the Last Interglacial and future Arctic energy budgets simulated by the IPSL-CM6A-LR model. Based on future climate projections, near-surface air temperatures are expected to rise in the Arctic throughout the year with a maximum anomaly relative to the pre-industrial period reached in winter. The annual evolution of the near-surface air temperature anomaly is different at 127 ka due to the seasonality of the insolation forcing: we have shown that Arctic climate warms in summer and autumn but cools slightly in winter and spring. Regarding mechanisms and feedbacks driving the Arctic warming, we can expect that longwave radiation plays a greater part in future Arctic amplification compared to the Last Interglacial due to the increased greenhouse gases forcing. This will actually be the topic of further work, which is not fully finalised at the moment.

It seems that in Otto-Bliesner et al. (2021), the winter sea-ice area is increasing for the IPSL-CM6A-LR model while in the current manuscript a decrease is presented. Please clarify this difference.

Yes, this is due to our definition of the Arctic region. We have averaged sea ice area between 60 and 90°N, thus excluding regions where sea ice anomalies are positive in winter i.e. south of Greenland and in the Sea of Okhotsk (fig. 7e).

Lines 108-114: two phases of the lig127k experiment are mentioned, one 350 years long and another one 550 years long. What is the difference between these two? Try to explain this more clearly.

We modified the text as follow:
"The model was first run for 350 years. This initial step constitutes the spin-up period, during which the model reaches a statistical equilibrium under the Last Interglacial forcing. The final state of this spin-up phase is used to initialise the reference PMIP4-CMIP6 *lig127k* simulation which has been run for 550 years. The last 50 years have been produced saving high-frequency outputs for the analyses of extremes or to provide the boundary conditions for future regional simulations."

*Lines 153-156: a cold bias over Greenland in the PI simulation does not need to correspond to a too small temperature increase due to the 127k forcings. The relationship between biases in the present-day climate and the response of a model to a given forcing is often far from straightforward. The authors nicely show and discuss this in the discussion section, but these lines seem to indicate otherwise. Please clarify.*

We fully agree with the reviewer and propose to remove the following sentences, since this is a topic which we go back to in the discussion section:
"This bias has already been identified in the evaluation of the IPSL-CM6A-LR present day climate. It could be slightly amplified by the prescription of a modern Greenland ice sheet in the LIG simulation."

*Lines 226-227: The authors mention that the drift in deep ocean temperatures are not negligible, however, they are not discussed at any point later in the manuscript. Please clarify.*

We did mention a drift in deep ocean temperatures, in order to explain the fact that the annual mean value of ocean heat storage is not zero. Figure 5 shows that indeed there is a small drift in ocean heat content. However, this figure was somewhat misleading because of the chosen vertical scale, and the fact that the regression was computed on the whole simulation. We propose to replace Figure 5 by the figure below, which shows that during the first 300 years, the ocean heat content undergoes significant oscillations, which decrease in amplitude during the last 200 years of the simulation, which are analysed in the present work. The trend computed over these last 200 years is small, but because of small oscillations in ocean heat content, it is nonetheless important to keep the last 200 years for the analysis. In fact, we do not discuss the influence of such a drift because we are mostly interested in changes in the upper layers of the ocean where air-sea exchanges take place. Figure 8 shows that the major oceanic temperature variations occur in the first 60 m of the ocean and do not reach the deepest layer of the ocean due to ocean stratification.
We therefore propose to replace the paragraph including lines 226-227 by the following text:
"The annual value of storage terms should be zero in the ideal case of an equilibrium climate. This is not the case for both simulations. The PI AHS and OHS are lower than the current observed energy imbalance of 0.5 W m−2 in terms of absolute value (*Roemmich et al., 2015*; *Hobbs et al., 2016*). However, the LIG AHS and more specifically the LIG OHS are far above this reference value since they are respectively equal to 0.5 and 1.1 W m−2. This "energy excess" may arise from assumptions made for the energy budget computation or from an ocean drift in the LIG simulation. Figure 5 shows that the SST and ocean heat content drifts are small over the last 200 years of the simulations."

Since most of the changes of interest later in our study occur in the first 60 metres of the ocean (cf. Fig. 8), we do not discuss these drifts further.

[Figure]

*Figure 1 : Time series of (a) the sea surface temperature (◦C) and (b) the vertically integrated ocean heat content (J m⁻²) averaged over the Arctic region (60–90°N). The time axis indicates the number of months since the year 1850. The black lines show the results of linear regressions over the last 200 years of the simulation.*

*Line 239: Are these atmospheric surface air temperatures, SSTs? Please clarify.*

These are near-surface air temperatures. This has been corrected.

*Line 242: Comparing the simulated Arctic LIG warming with other PMIP3/4 models seems more relevant then global means in light of the topic of this manuscript.*

We agree with this comment and we modified the text as follows:
"Change in insolation between the LIG and the PI periods leads to  an annual Arctic near-surface air temperature anomaly of 0.9°C. This value is in the range of the PMIP4 multi-model mean of 0.82 ± 1.20°C (*Otto-Bliesner et al., 2021*)."

Note that we made a mistake line 242: the annual Arctic near-surface air temperature anomaly is 0.9°C compared to PI and not 1.8°C.

*Lines 253-255: Clarify if these are regional, ocean or continental averages.*

Here, these are regional averages. We will modify the text accordingly.

***Technical comments:***

We thank the reviewer for his corrections. We will take them into account and correct the text accordingly.

*Line 14: radiation instead of radiations*
We replaced "radiations" with "radiation" whenever this term appears in the text.

*Lines 26 and 28: too many brackets?*
Yes, this is now corrected.

*Line 46: another instead of an other*
We replaced "an other" with "another".

*Line 98: check brackets*
We removed the extra parenthesis.

*Line 119: To prevent such so-called "paleo-calendar effects"….*
We modified the text accordingly.

*Line 123: cloud*
We replaced "clouds" with "cloud".

*Line 141: but it does not reproduce*

*Line 153: saptial is spatial?*
This is a typing error. We replaced "*saptial*" with "spatial".

*Line 266: One bracket too many*
We removed the extra parenthesis.

*Line 271: Be careful with the usage of 'near surface air', 'surface air', these things normally have different meanings. Clarify ones what you mean with it and consistently use the term thereafter.*
As we use t2m-temperature we will use near-surface air temperature hereafter.

*Line 306: is advected way from the Arctic basin; or advect out of the Arctic basin*
We replaced "is advected outside the Arctic basin to balance" with "is advecetd way from of the Arctic basin".

*Figures 4 & 5: clarify if these results are for the PI or LIG simulations.*
Figure 4, results are for the pre-industrial period. Figure 5, results are for the Last Interglacial. We modified the caption accordingly.
   Figure 4: "Annual zonal mean northward heat transport (PW) for the pre-industrial period. The northward heat transported computed as residual are represented by a solid line. The atmospheric heat transport is in black and the oceanic transport is in blue. The oceanic heat transport simulated by the IPSL-CM6A-LR model is represented by the dashed blue line."
   Figure 5: "Time series of (a) the sea surface temperature (°C) and (b) the vertically integrated ocean heat content (J m$^{-2}$) averaged over the Arctic region (60-90°N) for the Last Interglacial. The time axis indicates the number of months since the year 1850."

*Figure 8: these are all temperature anomalies?*
Yes. The first panel represents the atmospheric temperature anomaly. The second one shows the oceanic temperature anomaly.

References:

Hobbs, W., Palmer, M. D., and Monselesan, D.: An Energy Conservation Analysis of Ocean Drift in the CMIP5 Global Coupled Models, J.Climate, 29, 1639–1653, https://doi.org/https://doi.org/10.1175/JCLI-D-15-0477.1, 201

Kageyama, M., Sime, L. C., Sicard, M., Guarino, M.-V., de Vernal, A., Schroeder, D., Stein, R., Malmierca-Vallet, I., Abe-Ouchi, A., Bitz, C., Braconnot, P., Brady, E., Chamberlain, M. A., Feltham, D., Guo, C., Lohmann, G., Meissner, K., Menviel, L., Morozova, P., Nisancioglu, K. H., Otto-Bliesner, B., O'ishi, R., Sherriff-Tadano, S., Stroeve, J., Shi, X., Sun, B., Volodin, E., Yeung, N., Zhang, Q., Zhang, Z., and Ziehn, T.: A multi-model CMIP6 study of Arctic sea ice at 127 ka: Sea ice data compilation and model differences, Clim. Past, 17, 37–62, https://doi.org/https://doi.org/10.5194/cp-17-37-2021, 2021.

Otto-Bliesner, B. L., Brady, E. C., Zhao, A., Brierley, C., Axford, Y., Capron, E., Govin, A., Hoffman, J., Isaacs, E., Kageyama, M., Scussolini, P., Tzedakis, P. C.,Williams, C.,Wolff, E., Abe-Ouchi, A., Braconnot, P., Ramos Buarque, S., Cao, J., de Vernal, A., Guarino, M. V., Guo, C., LeGrande, A. N., Lohmann, G., Meissner, K., Menviel, L., Nisancioglu, K., O'ishi, R., Salas Y Melia, D., Shi, X., Sicard, M., Sime, 655 L., Tomas, R., Volodin, E., Yeung, N., Zhang, Q., Zhang, Z., and Zheng, W.: Large-scale features of Last Interglacial climate: Results from evaluating the lig127k simulations for CMIP6-PMIP4, Clim. Past, 17, 63–94, https://doi.org/https://doi.org/10.5194/cp-17-63-2021, 2021.

Roemmich, D., Church, J., Gilson, J., Monselesan, D., Sutton, P., and Wijffels, S.: Unabated planetary warming and its ocean structure since2006, Nat. Clim. Change, 5, 240–245, https://doi.org/https://doi-org.insu.bib.cnrs.fr/10.1038/nclimate2513, 2015.

---

## Author Comment (AC2)

**Response to RC2**

We would like to thank the reviewer for her/his careful reading and her/his comments about this manuscript. We hope that the revisions we plan to implement should satisfactorily address his comments.
Below, the reviewers' comments are highlighted in italic, and our responses follow in blue.

Marie Sicard on behalf of the co-authors
* * *
*General Comments*:

*This paper adds to the literature on the Last Interglacial, which has assumed newfound relevance as the modern climate continues to warm, while the Arctic heats up even faster. In this light, the paper is timely and based on a solid approach of diagnosing the Arctic warmth of the Last Interglacial in a climate model through the lens of energy changes and physical processes. That said, I come away underwhelmed by the purpose and significance of this study in its current form, because there is not a clear explanation of what new insights were gained. In other words, what did we learn about the Last Interglacial from this research that we didn't already know?*

In the revised version, we tried to emphasize the originality of this paper.

*The major takeaways about insolation forcing leading to large changes in sea ice, surface fluxes and time-lagged warming (e. g., peak warming during autumn, despite peak insolation forcing during summer) are consistent with findings from previous studies of the Last Interglacial and other orbitally warmed Arctic time periods such as the middle Holocene. However, that alone isn't grounds for an interesting new contribution. There may well be some new discoveries here that fill knowledge gaps, but those need to be articulated in the manuscript, especially since another recent paper on the Last Interglacial also reported changes in the Arctic energy balance and cloud responses using a different GCM (Guarino et al. 2020, Nature Climate Change). For example, it would help if the manuscript described which components of the diagrams of climate processes and feedbacks in Figures 13 and 16 are new and important discoveries from this study.*

The IPSL-CM6A-LR model shows a very different behaviour than the HadGEM3 model in terms of sea ice area variations and atmospheric energy balance (*Kageyama et al., 2021*). Moreover, *Guarino et al., 2020* only investigate changes in the atmospheric part of the energy budget. Here, we go further in the analysis by quantifying changes in the coupled atmosphere-ocean-sea ice Arctic energy budget. To our knowledge, this is the first time such an analysis has been carried out for the Last Interglacial.
Climate processes and feedbacks shown in figures 13 and 16 are not really new, since they have been identified as drivers of the future Arctic amplification, but they are important discoveries to explain the Last Interglacial Arctic warming.

**Specific Comments**:

*The Introduction needs a better motivation for the present study. This section contains a lot of general information about Arctic amplification and some background about the Last Interglacial climate, but there is no information given about what the numerous past studies of this time period have revealed*

*about the causes of the pronounced Arctic warming and what knowledge gaps the present study will fill. To first order, the enhanced Arctic warming is simply a consequence of much greater warm-season insolation, so there is no mystery. I suggest a structure along the lines of, "These previous studies of the Last Interglacial suggest <X, Y, and Z> as the major factors for the enhanced Arctic warming, but they were limited by <omissions or weaknesses in these past studies>. To better understand the physical mechanisms responsible for the dramatic Arctic warmth of the Last Interglacial, we are using a  climate model to diagnose the regional energy budget during this time period." A helpful framework is the "And, but, therefore" statement popularized by Randy Olson ([https://www.sesync.org/for-you/communications/toolkit/and-but-therefore-statement](https://www.sesync.org/for-you/communications/toolkit/and-but-therefore-statement)).*

We propose adding the following paragraph in the introduction (before line 73):
"Using the Earth system model of intermediate complexity MoBidiC, *Crucifix and Loutre (2002)* suggested that the precession is the main driver of annual mean temperature variations in the high latitudes of the Northern Hemisphere during the Last Interglacial. Through significant variations in summer snow cover, sea ice area and vegetation distribution, changes in summer precession modulate the surface albedo and then, explain most of climatic fluctuations in this region. However, while Crucifix and Loutre (2002) have shown that the thermohaline circulation has a limited impact on the high latitude climate, *Pedersen et al. (2017)* attributed changes in surface temperature to an increase in the annual mean strength of the overturning circulation from 15.8 Sv during the pre-industrial period to 21.6 Sv at 125 ka simulated by the high resolution EC-Earth general circulation model. Recently, *Guarino et al. (2020)* estimated surface heat balance over the Arctic Ocean with the HadGEM3 general circulation model to evaluate the link between Arctic warming and loss of summer sea ice at 127 ka. The authors found a positive anomaly of the net shortwave radiation at the surface, mostly caused by a substantial decrease of surface albedo. Compared with other CMIP6 models, HadGEM3 shows an unusual behaviour in terms of energy budget (*Kageyama et al., 2021*). The albedo feedback is strongly amplified in this particular model because of the significant summer sea ice retreat. It can be explained by the explicit representation of melt ponds in the CICE-GSI8 sea ice model which favours sea ice melt (*Flocco et al, 2012*).
These results have to be treated with caution since calendar has not been adjusted: the comparison with data and modern simulations at the seasonal time scale could be distorted (*Kutzbach and Gallimore, 1988*; *Joussaume and Braconnot, 1997*; *Bartlein and Shafer, 2019*). Moreover, previous studies did not clearly quantify the influence of each climate system components i.e. ocean, atmosphere, sea ice and continents, and their mutual interactions that contribute to the Last Interglacial Arctic warming. The aim of this study is  to better constrain their relative role based on an energy budget framework. To address this issue, we use the outputs of the IPSL-CM6A-LR global climate model to compare Arctic energy budget  during the Last Interglacial and pre-industrial periods "

We also suggest adding in section 3.22 (line 347) the following sentence:
"This result is in line with *Bakker et al (2014)* who identified a land-ocean temperature contrast in the mid-to-high latitudes of the Northern Hemisphere of 1.8 during the warmest months of the Last Interglacial (123-116.2 ka)."

*Line 87: Please include the latitude/longitude resolution of the model in degrees*

The horizontal resolution of the atmospheric model is 144x143 points in longitude and latitude corresponding to a resolution of 2.5°x1.3°.

*Section 2.1/2.2: There should be more justification for using this particular climate model in the study, especially because this is considered a low-resolution model. The improvements to this model version described in Section 2.2 are helpful, but how well does it simulate global and/or Arctic climate compared with other models (say those in CMIP5)? Also, what are the implications and limitations of using a low-resolution model for this study, which investigates dynamical changes in the atmosphere and ocean that might depend on relatively small-scale features such as transient eddies?*

   The model used in our study was developed as the main IPSL model for Phase 6 of the Coupled Model Intercomparison Project (Boucher et al., 2020). Accordingly, it was used for the whole CMIP6-PMIP4 project. However, for the HighResMIP part of CMIP6, a high-resolution configuration of the atmospheric model (i.e. LMDZ6) was also designed with 512x361 points, corresponding to an isotropic resolution of 50 km at 45° (*Hourdin et al., 2020*). In *Hourdin et al. (2020),* figure 5 shows the zonal and annual averages of the longwave, shortwave and total cloud radiative effect, of the top of the atmosphere outgoing shortwave and longwave radiation, and surface precipitation rate both simulated by the low and high-resolution versions of the model. There are no significant differences between both versions of the model. Moreover, we also computed the surface heat budget and did not find any major discrepancy between low and high-resolution versions of the model (fig. 1). Therefore, the IPSL-CM6A-LR model seems well adapted to this study despite its low resolution.

[Figure]

*Figure 1: Zonal and annual mean of the Northern Hemisphere surface heat budget simulated by the low and high-resolution version of the IPSL model. Heat fluxes are in W m$^{-2}$.*

*The 500-year spin-up is apparently not long enough for AMOC to reach equilibrium in the Last Interglacial run. It also isn't long enough to accurately compute stable deep-ocean temperatures as part of the energy budget analysis (Figure 5), so how confident can we be in the energy budget calculations of the ocean? The deep ocean heat content drift is quite large.*

   We acknowledge that figure 5b is misleading. This is mainly due to the chosen scale of the y-axis. Indeed, the slope of the regression line is only 0,00017 x 109 J m-2 in the550-yearsimulation and fluctuations are even smaller over the last 200 years. (i.e. the last 2400-time steps of the simulation).

Figure 5a and 5b illustrate that the upper layers of the ocean stabilize faster than the deepest ones, but the ocean drift remains at a very low level.

In order to improve the clarity of these graphics, we have computed the regression line only for the last 200 years of the simulation and adjust the y-axis of the plot (see figure below).

[Figure]

*Figure 2: Time series of (a) the sea surface temperature (∘C) and (b) the vertically integrated ocean heat content (J m−2) averaged over theArctic region (60–90°N). The time axis indicates the number of months since the year 1850.*

*Figure 2: This figure needs a better explanation. For one thing, the caption should say that green symbols—not the green contour---show good model-proxy agreement (and state what is considered to be good agreement). Second, the overlain symbols show the paleodata as a point comparison with the model, but the caption implies that the entire maps are a proxy comparison with the model. I think the shadings on the maps are showing just the model-simulated difference between Last Interglacial and Pre-industrial climate, correct?*

Yes, we agree with the reviewer. We have modified the caption for Figure 2 accordingly:
"LIG-PI anomaly of the near-surface air temperature (°C) simulated by the IPSL-CM6A-LR model (color shading) and reconstructed from proxy data synthesis (filled markers) as published by Otto-Bliesner et al. (2021). Symbols represent the source of surface air reconstruction: circles for the compilation by Hoffman et al. (2017), squares for the compilation by Capron et al. (2014, 2017) and triangles for the Arctic compilation. Sites showing good model-data agreement (i.e. considering a data uncertainty of ± 1σ) are indicated by a green outline. "

*Line 165: Why was the energy budget calculated over the final 200 simulation years, rather than just the final 50 (as implied in line 112)?*

In line 112, we use the term high resolution to refer to daily resolution. For this study we do not need daily outputs because we analyse the Arctic energy budget as a seasonal average. We calculate the energy budget over the final 200 years to smooth out the inter-annual and decadal variability.

*Lines 239-241: Please discuss the large discrepancies in simulated global temperature anomalies at the Last Interglacial (near zero) versus the 2 K warming reported earlier (from proxy data?) in lines 55-56.*

The IPSL-CM6A-LR model simulates cooler mean annual near-surface air temperatures than the reconstructed value of 2°C because the 127 ka period is near the peak warmth but does not correspond to the warmest period of the Last Interglacial. Moreover, this mismatch can also be partly amplified by the lig127k experiment protocol itself because of the prescription of the ice sheet topography and vegetation to their modern state (*Otto-Bliesner et al., 2021*). Therefore, ice sheet-climate and vegetation-climate feedbacks are neglected. We have added this discussion line 242.

*Line 249: Should "anomaly" be changed for clarity to "anomalies" in reference to the separate magnitudes of the ocean and land temperature anomalies, both of which are larger during winter than spring in Figure 7 a,b?*

Yes, we rephrased it accordingly.

*Line 258-259: Is it correct to say that snow cover in Figure 7k doesn't respond to the summertime warming? Most places appear to show a decline compared with PI.*

Snow cover responds to the Last Interglacial summertime warming, since there is a large decrease in snow fraction from spring to summer (see figure 3 below). However, the summer snow fraction anomaly does not seem to be directly related to the near-surface air temperature anomaly pattern contrary to what is observed in autumn (see figures 7d and 7l of the manuscript): the near-surface air temperature increases globally over the continents compared to the pre-industrial period, while snow fraction decreases very locally. As mentioned in the manuscript, summer snow cover has almost completely melted during the pre-industrial period (see figure 3 below), which explains why changes in snow cover are so small.

[Figure]

*Figure 3: Seasonal cycle of the snow fraction for the pre-industrial period (top) and the Last Interglacial (bottom).*

*Lines 303-304: Why does only the upper ocean warm during summer (top 100 m)?*

During summer, the oceanic upper layers warm because they absorb more shortwave radiation. The excess heat does not reach the deep ocean because of the ocean stratification that limits the depth of seasonal heat exchange and mixing.

We modify the text accordingly:
"The surface heat budget over the ocean confirms that the upper layer of the ocean warms up during the LIG. Unlike the atmospheric warming that affect the entire atmospheric column, the increase in ocean temperatures only appears in the upper 100 m of the ocean (fig. 8). This can be explained by the ocean stratification that limits the depth of seasonal heat exchange and mixing with the deepest oceanic layers."

*Line 329: The latent heat flux has to be directly correlated with evaporation, because that flux is the evaporation rate times the latent heat of vaporization.*

Yes, we agree with the reviewer and removed the sentence from the text.

*Line 342: Could you briefly explain what Bjerknes theorized regarding heat transports, rather than just pointing readers to an indirect reference?*

We modified the text as follows:
"Since it decreases from PI to LIG, the AHT anomaly almost balances the OHT anomaly. This strong negative relationship between changes in AHT and OHT was first suggested by *Bjerknes (1964)* and has been simulated by many modelling studies (see *Swingedouw et al., 2009* for example). "

*Lines 377-378: What is the physical explanation for the contribution of potential energy storage and its seasonal dependence? The text states that the change in potential energy storage follows the same seasonal variation as the internal energy storage, which is clearly temperature-dependent. But Table 2 shows differences between these two components, such that internal energy storage increases equally during spring and summer in LIG, whereas potential energy storage has a distinct spring peak.*

By definition, the seasonal storage of potential energy depends on the seasonal variations of the geopotential. Figure 4 below shows the annual evolution of the geopotential averaged over the Arctic (60 °N-90°N) as a function of pressure.

[Figure]

*Figure 4: Annual evolution of the geopotential (m² s⁻²) averaged over the Arctic region (60-90°N) as a function of pressure.*

    The largest geopotential anomalies occur at the top of the atmosphere from May to October when the air temperature anomaly is the largest. However, the geopotential anomaly does not seem to be associated to the significant changes in temperature observed between 600 hPa and 300 hPa in summer (see fig. 8 in the manuscript).

    In line 377-378, we are only focused on changes in potential energy storage between summer and autumn. We now specify on the text that the term "trend" is related to the trend from summer to autumn. We therefore modified the text as follows:

"The anomaly of the internal energy storage depends on air temperature fluctuations from one season to the other. As illustrated in figure 8, the air temperature increases from summer to autumn near the surface but over the rest of the atmospheric column, the air temperature peaks in August. The potential energy storage anomaly is also strongly dependent on the temperature in the atmospheric column and follows the same trend as the internal energy storage from summer to autumn."

*Line 384: How does Figure 8 show that feedbacks operate in the lower atmosphere but not above, given that the maximum atmospheric warming in that figure occurs aloft in the 300-600 hPa layer?*

    In summer, the maximum atmospheric warming clearly occurs in the 300-600 hPa layer. However, the warming is restricted to the lower layers of the atmosphere during the autumn.

***Technical Corrections:***

*A thorough proofreading is needed.  I identified a few typos or misspellings below but stopped keeping track.*
We thank the reviewer for his corrections. We will reread the manuscript carefully.

*"radiation", rather than "radiations"*
We replaced "radiations" with "radiation" whenever this term appears in the text.

*Line 30: should be "have" been*
Yes, we replaced "has" with "have".

*Line 43: clarify what's meant by "blocks longwave radiation" (trapping longwave emission from the surface?)*

Yes, we modified the text accordingly.

*Line 46: Should be "another" process*

Yes, we replaced "an other" with "another".

*Lines 93-94: I don't understand what this sentence means ("which implies that event through vegetation type. . . ").*

Here, we made a typing error. The correct sentence is: "Finally, the ORCHIDEE model also includes a carbon cycle representation, which implies that event though vegetation types are prescribed in each grid box the seasonal evolution of the leaf area index is computed."

*Line 101: typo*

This has been corrected.

*Line 108: "Interglacial"*

We replaced "Interglacials" with "Interglacial".

*Line 110: typo*

We replaced "longenough" with "long enough".

*Line 141: typo*

This has been corrected.

*Line 209: Should the arrow for Fbot in Figure 3 be reversed, since the text says that Fbot cools sea ice when positive?*

Yes, we modified the figure.

*Figure 4: State in the caption that this result applies to the pre-industrial simulation. Likewise, state that Figure 5 refers to the Last Interglacial simulation.*

We modified the caption accordingly.

[revised manuscript text omitted]

---

## Author Response (AR2)

We would like to thank the reviewers for their proofreading during this second revision phase. We have modified the manuscript according to the reviewer's comments.

Marie Sicard on behalf of the co-author